# Aerosol dynamics and dispersion of radioactive particles

Pontus von Schoenberg[1,2], Peter Tunved[2,3], Håkan Grahn[1], Alfred Wiedensohler[4], Radovan Krejci[2,3], Niklas Brännström[1]

[1] Department of CBRN Defence and Security, The Swedish Defence Research Agency, FOI, SE-90182 Umeå, Sweden
[2] Department of Environmental Science and Analytical Chemistry (ACES), Stockholm University, 106 91 Stockholm, Sweden
[3] Bolin Centre for Climate Research, Stockholm University, S 106 91 Stockholm, Sweden.
[4] Leibniz Institute for Tropospheric Research (TROPOS)

*Correspondence to*: Pontus von Schoenberg (pontus.von.schoenberg@foi.se)

**Abstract.** In an event of a nuclear power plant failure with release of radioactive material into the atmosphere, dispersion modelling is used to understand, how the released radioactivity is spread. For the dispersion of particles, Lagrangian Particle Dispersion Models, LPDMs are commonly used in which model particles, representing the released material, are transported through the atmosphere. These model particles are usually inert and undergo only first order processes such as dry deposition and simplified wet deposition along the path through the atmosphere. Aerosol dynamic processes including coagulation, condensational growth, chemical interactions, formation of new particles and interaction with new aerosol sources are usually neglected in such models. The objective for this study is to analyse the impact of these advanced aerosol dynamic processes if they were to be included in LPDM simulations for the use in radioactive preparedness. In this investigation, a fictitious nuclear power plant failure is studied for three geographically and atmospherically different sites. The incident was simulated with a Lagrangian single trajectory box model with a new simulation for each hour throughout a year to capture seasonal variability of meteorology and variation in the ambient aerosol. We conclude that: a) modelling of wet deposition by incorporating an advanced cloud parameterisation is advisable since, it significantly influence simulated levels of airborne and deposited activity including radioactive hotspots, and b) we show that inclusion of detailed ambient aerosol dynamics can play a large role for the model result in simulations that adopt a more detailed representation of aerosol cloud interactions. The results highlights a potential necessity for implementation of more detailed representation of general aerosol dynamic processes into LPDMs in order to cover the full range possible environmental characteristics that can apply during a release of radionuclides into the atmosphere.

## 1 Introduction

Atmospheric dispersion models are used to simulate how various kinds of pollutants disperse in the atmosphere. Dispersion models for emergency preparedness and more specifically Lagrangian Particle Dispersion Models (LPDMs) currently have very simplified descriptions of the complex aerosol dynamic processes that transform both the particle number size distribution (PNSD) and the chemical composition The purpose of this study is to investigate whether detailed aerosol micro-physical processes are important when simulating the transport and deposition patterns following a release of radionuclides from a

nuclear power plant failure. The aim of the study is to investigate the potential effects that could result from inclusion of

detailed aerosol microphysics in dispersion models. Can these processes change simulated aerosol lifetime and deposition fields and are these effects of a high enough degree to encourage implementation these process descriptions into the framework of currently adopted dispersion modelling techniques?

Release of radiological species in the atmosphere can subsequently have a great impact on humans, the environment and its ecosystems (IAEA, 2006). In radioactive emergency preparedness, the dispersion modelling results are part of the decision

support. In the early phase of an emergency, output from dispersion modelling may be the only source of information readily available to the decision maker. European directives are stated in (Council of the European Union, 2013) and many member states have additional regulations. Radioactive particles can cause harm through internal or external exposure (ICRP, 2007). Internal dose comes from inhaled particles or through digestion. The size and hygroscopicity of the particles are strong determinants for where in the respiratory system the particles deposit when inhaled, which in turn determines the internal dose

and the severity of the potential injury. The current dose models used in radiological preparedness are described by the International Commission on Radiological Protection, ICRP (ICRP, 1994). Thus, the size of the particle carrying the nuclides is a property of integral importance in estimating direct as well as delayed exposure. The uptake of radioactive isotopes in vegetation depends also on size and chemical composition of the aerosol particles carrying isotopes (IAEA, 2009).

By definition, an aerosol consists of particles or droplets suspended in a gas phase of varying complexity. Aerosol particles,

both natural and anthropogenic, are either directly emitted or formed in situ in the atmosphere through gas-to-particle conversion (nucleation). Furthermore, aerosol particles are highly variable in size, from nm to µm in diameter. The majority of the particle number concentration appears in the ultrafine size range below 100 nm (often divided into *nucleation mode* with diameters up to 10 nm and *Aitken mode* about 10 nm to 100 nm in diameter). The surface area and the mass concentration are often highest in the *accumulation mode* size range, typically between 100 nm to 1 µm in diameter. However depending on

location and season, a substantial part of the aerosol mass and surface can be found in the coarse mode. Coarse mode particles typically have a short lifetime in the atmosphere due to their rapid settling velocity. Particles larger than the accumulation mode are denoted the course mode (see naming conventions in Fig. 1). The PNSD, chemical composition and hygroscopicity are critical parameters determining the fate of the particles during transport in the atmosphere. These are closely linked to key processes such as condensation, coagulation, dry and deposition including the potential to act as cloud condensation nuclei,

CCN.

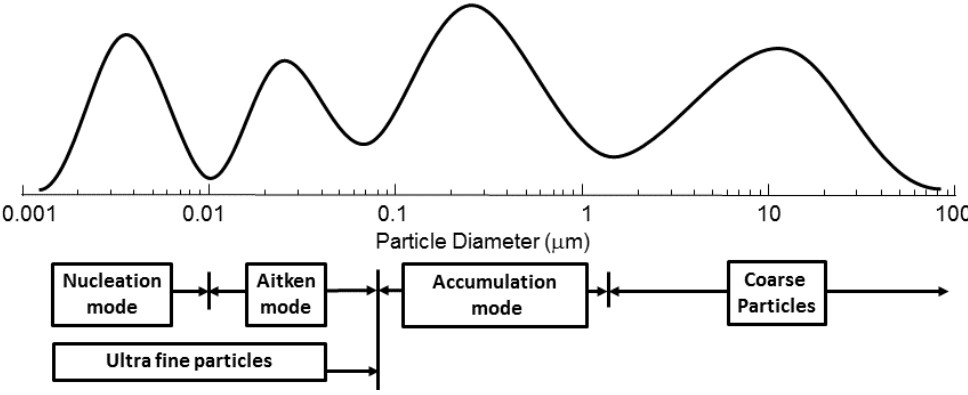

**Figure 1: An example of a particle size distribution consisting of a number of different modes. The names of the different modes and sizes are shown.**

Condensation of low volatile gas phase species and evaporation from the surface of the particles are continuous mechanisms that affects the size of the aerosols and the chemical composition. Coagulation, when particles agglomerate to form larger particles, reduces the number concentration of the particles and transform the size distribution while conserving the mass of the species within the aerosol**.**

Aerosol particles are removed from the atmosphere by dry and wet deposition. Dry deposition efficiency is strongly size dependent and ultrafine particles and super micron particles have comparably short lifetime compared to accumulation mode aerosol.

The high deposition velocity due to Brownian motion is most effective on particles smaller than 50 nm, resulting in a comparably high deposition velocity. Super micron particles, in the coarse mode, Fig. 1, have a relatively short lifetime (high deposition velocity), due to gravitational settling and inertial impaction. The accumulation mode,  represent the size range of aerosols with the longest lifetime in the atmosphere, and thus how rapidly a trace substance enter the accumulation mode through either condensation or coagulation is of crucial importance for the atmospheric lifetime of the trace substance in the atmosphere.

Coagulation, condensation, chemical interactions with surrounding gases and hygroscopic growth determine the composition of atmospheric aerosols. Size and chemistry of aerosol determines cloud forming potential (acting as CCN) and thus efficiency of wet removal. Wet removal through in-cloud and below-cloud scavenging is the most important sink for atmospheric aerosols. There are many good sources for in depth information concerning aerosol physiochemical processes and atmospheric aerosols e.g. (Seinfeld and Pandis, 2006).

The deposition field, defined as the amount and spatial distribution of particles deposited to the surface, determines the external dose to people and environment. When particles deposit, they act as a sink for the air concentration as well as a "source" for the deposition field. In radiological emergency preparedness both the deposition field and the remaining air concentration are of importance for calculating external dose and inhaled internal dose.

Ideally, all atmospheric aerosol processes should be considered in dispersion models. Due to computational limitations and the complexity of physical parameterisations, dispersion models for emergency preparedness have been developed based on simplified physics. When it comes to Lagrangian Particle Dispersion Models (LPDMs) that rely on the transport of discrete model particles, only first order processes can be applied with computational ease. The rate by which coagulation, condensation, dry deposition, cloud formation, precipitation and wet deposition affect the aerosol particle population depends on the dynamic process considered in the air parcel. Traditionally, the aerosol dynamic is not simulated in LPDMs to this extent. Examples of LPDMs are the open source model FLEXPART (Stohl et al., 1998;Pisso et al., 2019), the UK Met Office dispersion model NAME (Jones et al., 2007) or the NOAAs HYSPLIT model (Stein et al., 2015). They all have different parametrizations for wet and dry deposition. The released model particles in a LPDM usually remain in the same size class as prescribed by the sources function for the full duration of the simulation, or until they are lost through deposition processes. Treating the model particles as discrete entities without any interaction with surrounding aerosol particle population introduces errors in the simulations of both dry and wet deposition processes, which are strongly dependent on the PNSD and abundance of both scavenged and surrounding aerosol particles. This treatment further neglect condensational growth and coagulation into a size where they can act as CCN and be available for in-cloud scavenging. Therefore, the total wet deposition could be underestimated substantially as there are no dynamic feedbacks on the aerosol-cloud interaction.

This dynamic feedback can be viewed from two different angles linking to competitive growth during cloud formation on the one hand, and condensation growth on the other hand. As wet deposition to a substantial degree occurs through nucleation scavenging, parts of the available CCN's are removed. If the PNSD remains static, a second cloud cycle will tend to result in activation starting at a lower size range, i.e. the activation and subsequent removal "eats" its way through the distribution taking chunks away from right-to-left with increasingly lower activation radius as a result. This will make smaller and smaller particles available for in-cloud scavenging and removal. Now, if the activation diameter instead would be fixed, and for simplicity assuming it to be 100 nm, once all 100 nm particles are removed no more in-cloud scavenging can occur. This of course is unrealistic as it is well established that activation is controlled by competitive growth, and the lower number of large particles, the smaller activation radius will result for a given set of conditions. Hence, the cloud activation radius and removal will be adjusted based on the previous removal events. The second link is when particles are grown due to condensation growth (and to lesser extent coagulation). These processes bring particles that otherwise would be too small to be activated into a size range where they in fact may become actual CCN's. Studying this dynamical coupling (or feedback) between growth, removal and cloud droplet activation is one of the main targets of this study.

There are Eulerian methods of dispersion modelling which have a more advanced take on aerosol dynamics, for example WRF-Chem (Grell et al., 2005) has an optional aerosol module simulating aerosol nucleation, condensation and coagulation. WRF-Chem is an online model where aerosol microphysical processes are simulated within the meteorological model itself. Simulating aerosol dynamics on a regional to global scale is also done in General Circulation models, e.g. (Vignati, Wilson et al. 2004), Chemical transport models, e.g. (Spracklen et al., 2005) and air quality models, e.g. (Zhang et al., 1999). There are examples of Eulerian-Lagrangian hybrid models, for example in (Danielache et al., 2019) where the Eulerian part calculated

the chemical transformation and the Lagrangian was used for transport. For emergency preparedness fast models are essential. To our knowledge there has been no solution presented providing sufficiently fast and detailed aerosol dynamics representations within the strictly Lagrangian framework applied in LPDM's used for dispersion simulations of radionuclides in a Fukushima type of accident, dispersion of emitted $^{137}$Cs is of major concern, since it is one of the isotopes giving long-term complications. $^{137}$Cs is therefore continuously monitored around the globe (IAEA, 2010). A release of vaporised $^{137}$Cs

condenses on the surface of the surrounding aerosol particles giving it an activity size distribution largely following the surface area size distribution of that carrier aerosol. In a LPDM, the dispersion of the aerosol is simulated by transporting model particles that each represents a part of the source. Henceforth, the individual particles will only undergo first order dynamics (i.e. highly parameterized wet removal, dry deposition and radioactive decay). Neither the PNSD, nor the chemical composition of the advected particle will change in the classic way as LPDM simulations are performed. This potentially pose

a significant problem. As previously described, removal through wet deposition is highly dependent on both the PNSD and the chemical composition of the aerosol particles. Without altering processes of the particle population, the simulation is prone to the risk of either overestimating or underestimating the removal depending on the initial conditions prescribed at the release point.

    In this study, a Lagrangian trajectory model Chemical and Aerosol Lagrangian Model, CALM (Tunved et al., 2010) was used.

CALM simulates the evolution of the physical and chemical properties of an aerosol following an air mass trajectory. The air mass trajectories are calculated using meteorology from a numerical weather prediction model to describe the path of the trajectory and its meteorological properties. CALM calculates the transformation of the PNSD and associated size resolved chemical composition due to nucleation, coagulation, condensational growth, dry and wet deposition, chemical transformation and new sources along the trajectory. In this study,  CALM have been modified (from a coupled two box model, simulating

the mixing layer and the residual layer) into tracking one box forward in time that moves also in height above ground in and out of the mixing layer (depending on the current trajectory). In this way, the used model setup does not simulate a single particle, or a static number size distribution, i.e. the most commonly used designs in classical LPDM models. Instead we allow the simulated trajectories to carry a complete PNSD that will age and transform during transport. Simulations have been done for different sites and under different meteorological conditions to account for the variations that can occur due to different

weather conditions, different air masses and new aerosol sources along the path. They have all been initialised with a measured PNSD unique for that time and location. One should bear in mind that the purpose of this study is not to calculate atmospheric dispersion (the single trajectory approach is not suited for dispersion calculations) but rather to study the potential impact resulting from omission of a detailed aerosol dynamic treatment, including the activation into cloud droplets, in current LPDM frameworks.

In this experiment however, the wet deposition scheme is more advanced than in many LPDMs using a scavenging coefficient. It includes below-cloud scavenging and in-cloud scavenging with CCN activation based on updrafts and to atmospheric lifetime and deposition efficiency of radioactive material compared to the standard way currently adopted in LPDMs i.e. treating deposition as a first order process only."

The purpose of the study is to investigate:

- When is a more sophisticated aerosol dynamics treatment in LPDM simulations needed?
- Is the current treatment of aerosol removal processes in LPDMs sufficient for the simulation used in radiological emergency preparedness?
- Which aerosol dynamic and cloud processes are most crucial to include in LPDMs to improve their accuracy?
- How do geography-specific and seasonal characteristics of meteorology and source profiles influence the
atmospheric lifetime of radionuclides emitted from an intermittent point-source?

## 2 Method

To examine the impact of aerosol dynamics in a LPDM framework, the Lagrangian trajectory model CALM was used (Tunved et al., 2010). Using CALM, the evolution of an aerosol was simulated along a large set of air parcel trajectories originating from three different measurement stations in Europe.

A full year of PNSD measurements of the ambient aerosol at each the three stations initiated the model for each individual simulation. Different modelling experiments were performed for all the trajectories to investigate the impact of simulating different processes.

### 2.1 Trajectory calculations

By simulating 24 different trajectories each day (one for each hour) the variability of the ambient PNSD, in the meteorology
and transport paths could be analysed. An air mass forward trajectory by definition describes the advection of an infinitesimal particle and can be used to describe the transport of a released substance or an air parcel. The three dimensional (latitude, longitude and height) trajectories used in this study were calculated with the model HYSPLIT in forward mode (Stein et al., 2015) using fields from the meteorological model GDAS (NOAA 2019) with a geographical resolution of 1 degree in latitude and longitude and temporal resolution of 3 hours. From these calculations, both the trajectory path (based on 3-dimensional
mean winds) and important meteorological parameters along the path was retrieved and used as input for CALM. In each individual simulation, CALM was run along the airmass trajectory for the duration of 10 days. In theory, there would be 365*24=8760 trajectories to analyse for each station for a full year (that is not a leap year) but due to missing data in the measurement series the actual number of trajectories is somewhat lower, see statistics in Table 1.

### 2.2 Measurement stations and aerosol data

The three chosen stations represent climatologically different regions and the trajectories from the stations travelled though different types of terrain. The ambient aerosol measurements at the stations were then used to initiate the CALM model. As the trajectories are calculated using re-analysed data, simulated travelled paths and associated meteorology will vary following seasonality and climatological characteristics in different environments as well as exposure to different source patterns. Together with the seasonal variability of the ambient PNSD, the simulations will cover a wide range of characteristics, which

will provide the basis for the evaluation of the impact of various processes affecting deposition fields and residence time of radionuclides attached to the aerosol phase.

The measurement station in Melpitz (Melpitz, 2020) in the centre of Europe was chosen as a representation of the Central European background with transport patterns mainly from Eastern and Western Europe. Neo in Greece (Neo, 2020), represents a coastal area where the ambient aerosol is strongly influenced by local sources by the coast and long range transport, in this

case from northern Italy and Balkans, occasionally with Saharan dust. The forward air mass trajectories starting at Neo often travels over the Mediterranean Sea. Zeppelin in the polar region (Zeppelin, 2020) represents a remote background station with both natural sources and long range transport from anthropogenic sources. At Zeppelin, the ambient background has a substantially lower particle number concentration then the other two stations. The location of the three stations can be seen in Fig. 2.

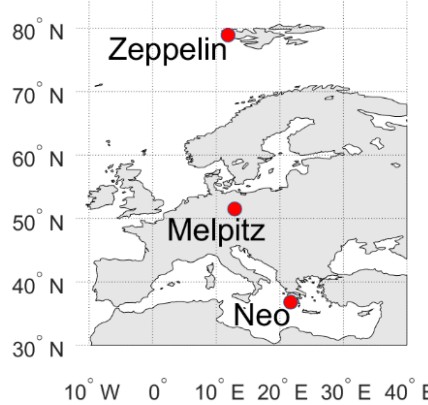


**Figure 2: Measurement stations. Particle number size distributions (PNSD) measured at the stations are used to initiate the trajectory calculations.**

The measurement time series should have as small gaps as possible for the chosen years (which should be near each other not to involve climatological differences). This made us chose different years for each station, still representing seasonal variability

for the analysis. For Melpitz the year 2008 was chosen where 7995 out of the years 8784 hours had measurements (91% coverage) (Reddington et al., 2011) for Neo the year 2012 with 7769 out of possible 8784 (88% coverage) and for Zeppelin 2010 with 7178 out of 8760 (82% coverage) (Tunved et al., 2013). For Zeppelin we have omitted the month of December since there were too few individual trajectories to analyse (based on the availability of measured size distribution).

All three stations observe PNSD using similar setups of different kinds of Mobility Particle Size Spectrometers, MPSS. Sizes

covered by the instruments range from a couple of nanometres up to several hundred nanometres. The super-micrometre size range is however not included in this study Doing so, we are aware that the omission of coarse mode potentially could introduce significant deviation from current results. This is especially true for occasions where the surface area is completely dominated by super-micron particles. Being prone to rapid dry deposition through sedimentation, similar situations could result in comparably fast removal of attached radionuclides.

Information about the instrumental set up can be found in e.g. (Tunved et al., 2013) for the Zeppelin site and (Birmili et al., 1999) for the Melpitz site. PNSDs observed at NEO is pending publication. The stations are part of the European Research Infrastructure ACTRIS in which the measurements and the quality assurance and control are harmonized (Wiedensohler et al., 2012;Wiedensohler et al., 2018).

In order to harmonize the data into a format that directly can be input to the model, each hourly PNSD was fitted under the
assumption that each PNSD can be represented by three log-normally distributed modes. Each mode is defined by a number concentration (N), a geometrical mean modal diameter (Dg) and a geometric standard deviation (GSD) that defines the spread of N around Dg for each mode. See Fig. 3 for PNSD (left) and Surface Size Distribution (right) for the three stations. The total number concentration and total surface area is in the bottom row. The best fit was found through solving a constrained minimization problem that gives the optimal distribution into three distinct log-normal modes under the constraint of non-
negative weights. In a few cases (0.6%) the automatic fitting process did not work and these occasions were not simulated. Seasonal variations in the PNSD can be noticed for all stations, where Zeppelin stands out, c.f. Fig. 3. PNSD seasonality in the artic is driven by both local sources, remote sources and transport patterns as well as differences in precipitation patterns, (Tunved et al., 2013). The total concentration of particles is lower in the arctic than for the other station which makes it more sensitive for variations. The spring period, February to April, consists of an aged and elevated accumulation mode aerosol
strongly influenced by remote sources linked to meteorological transport patterns and inversions trapping the aerosol and reducing dilution. This is commonly referred to as *Arctic Haze*. The number concentration during the summer period, May-September, is formed through mainly gas-to-particle conversion resulting in a pronounced Aitken mode towards as well as intermittent presence of a nucleation mode, c.f. Fig. 2. Since particle formation is dependent of sunlight, there are also strong diurnal variations in this period."


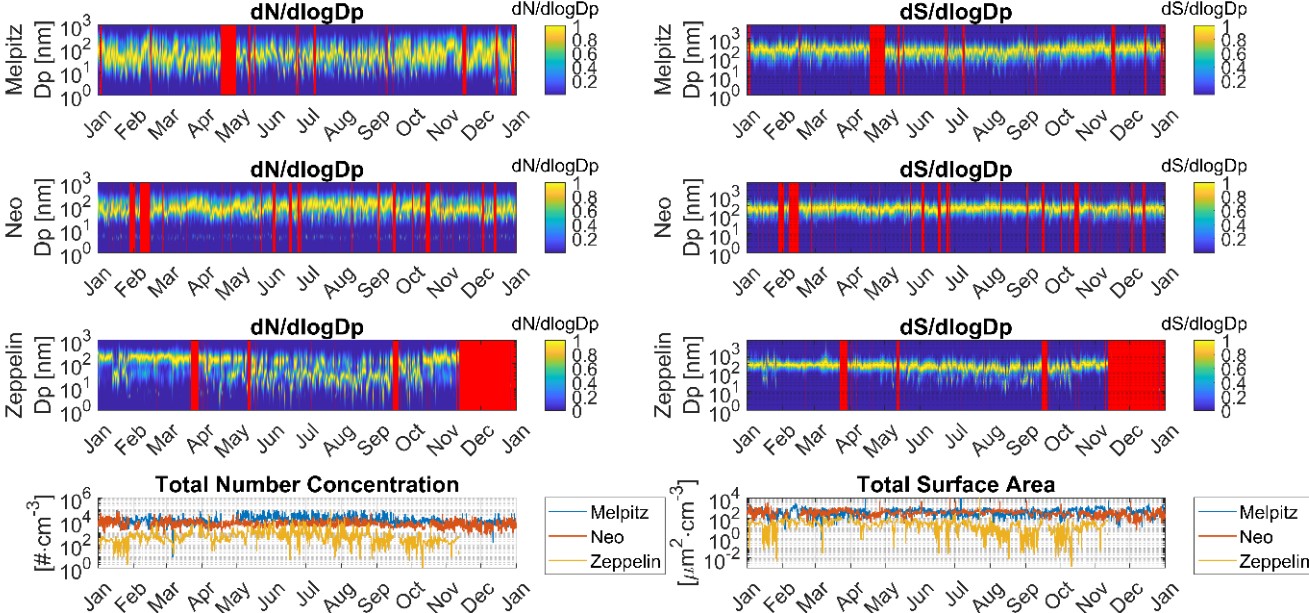

**Figure 3: Seasonal development of initial PNSD, normalised to the maximum in the bin with the highest concentration (left column) and Initial Surface Size Distribution, normalised to the maximum in the bin with the largest surface area (right column). Melpitz 2008 (top row), Neo 2010 (second row) and Zeppelin 2012 (third row). Three modal fit calculated from measurements at each station. The bottom row holds total number concentration (left) and total surface area (right).**

The aerosol particles are assumed to be internally mixed i.e. all aerosol particles of a certain size have the same chemical composition (as opposed to externally mixed, where one size class can contain particles of various composition). CALM divides the internal mix of the particles into three different chemical groups, ammonium bisulfate, condensable and partially water-soluble organic compounds and primarily emitted species. The setup assumes irreversible condensation of ammonium bisulfate and organic material, and no chemical reactions take place in the dry particle phase (although wet-phase oxidation of sulphur dioxide in cloud droplets is considered when present). In the current setup we have added a third condensable species with saturation vapour pressure set to zero, to replicate intermittent irreversible condensation of $^{137}$Cs on the size distribution. The initial aerosol chemical composition comes from a general description in (Putaud et al., 2004) where their definitions of a "Near City Urban" composition has been used to represent the Central European background for Melpitz and their definition of a "Natural Rural" composition has been used for representing the clean remote background for Zeppelin and the marine background for NEO, respectively, see composition in Table 1. Chemical transformation of the particles has not been studied and this crude generalisation was considered sufficient for this study.

### 2.3 Radioactive particles

Our topic of interest is the dispersion of radioactive particles from a nuclear power plant failure. $^{137}$Cs is monitored all around the globe for radiological preparedness since it is one of the isotopes causing most of the long-term damages. For this study,

we assume that all released $^{137}$Cs from a nuclear power plant failure is attached to the ambient aerosol in the vicinity of the power plant, as assumed in the study by (Kristiansen et al., 2016). This assumption will serve the purpose for this study, even though particles containing $^{137}$Cs internally (not only on the surface) have been found after the Fukushima accident (Adachi et al., 2013), (Higaki et al., 2017). The caesium is present in small enough quantities not to affect the aerosol dynamics in the model, but instead used as an inert tracer. In this context, the unit of Cs is arbitrary but we will refer to it as Becquerel throughout the paper. Throughout the simulation, the distribution of $^{137}$Cs and ambient aerosol respectively was traced, showing the effect of the aerosol dynamic processes.

In the experiments with CALM, the release of $^{137}$Cs had a duration of 800 sec, and started 60 min into the simulation. The $^{137}$Cs was released as a gas and the concentration for saturation above a flat surface was set to zero to make it condense irreversible on the surface of the particles present in the ambient aerosol. Throughout the simulation the distribution of $^{137}$Cs and the number distribution of the whole aerosol could be traced respectively, showing the effect of the aerosol dynamic processes.

## 2.4 Model setup

CALM describes the evolution of the PNSD as well as the $^{137}$Cs activity distribution. The aerosol dynamic processes: condensational growth, coagulation, nucleation of new particles, new sources, wet deposition and dry deposition were turned on or off in different experiments to analyse their individual impact.

The dry deposition includes Brownian diffusion, most effective for particles smaller than 50 nm, and gravitational settling and inertial impaction, which is most effective for particles larger than a few micrometres in size. Dry deposition in the model setup is only active when the box is in the mixing layer, when particles can reach the ground. This neglects the effect of gravitational settling for particles above the mixing layer and this effect can be large when aerosol surface is dominated by coarse mode particles.

The model design utilizes a hybrid approach and considers two different compartments: one for ambient aerosol dynamics and one for aerosol cloud interactions and in-cloud scavenging. The dry "ambient dynamic" box considers detailed descriptions of gas-to-particle-conversion, dry deposition, and coagulation. Decoupled from this box we run an adiabatic 1-dimensional cloud module that calculates activation and growth of aerosol particles in an ascending air parcel. The cloud model is run separate from the ambient dynamics box when clouds are prescribed. The cloud compartment results in a droplet distribution of the activated particles, which in turn allow us to calculate a liquid water content (LWC). The environmental parameters framing the cloud calculations are based on meteorological parameters provided by the meteorological model (GDAS) which are calculated every 3 h. Exactly how this is done is outlined below. Once formed, available $SO_2$ is equilibrated to the bulk water together with ozone and hydrogen peroxide. This allows for calculation of pH and concentration dependent liquid phase oxidation of sulfur dioxide. This means that if the cloud does not precipitate, the sulfate produced through this pathway is distributed over the activated particles. This means that the cloud does in fact have the potential to act as a source of aerosol mass.

Apparently, there is a discontinuity comparing on the one hand the cloud, and on the other hand the ambient box. We have
chosen this approach since we want to retain the key processes of activation and its link to aerosol size distribution properties
and chemistry. Once the cloud dissipates, the effect it has had on the aerosol in the ambient dynamics box (in-cloud chemistry
and in-cloud scavenging) is evaluated based on the fraction of the box that has been influenced by the cloud, which is
determined from humidity profiles and fractional cloud cover.

Concerning wet deposition, the process when the aerosol particles act as cloud condensation nuclei and initiate a cloud droplet
is called activation. CALM utilize a 0-dimensional activation scheme, where the activation and growth are explicitly calculated
in an ascending air parcel at prescribed, but variable, updrafts. This determines the smallest size of the activated particles at
each instance of cloud formation. Clouds are considered for every three hours, the model calculates a vertical profile  of
pressure, temperature, and humidity as well as fraction of low, midlevel, and high clouds. If clouds are present in the vertical
column, the model checks if the humidity at the altitude coinciding with that of the air parcel. The vertical resolution of the
meteorological model used (GDAS 1 degree) is, starting from bottom-up at 1000 hPa with 25 hPa resolution up to 900 hPa,
and then 50 hPa resolution up to levels relevant for the current simulations. The vertical resolution is 1 degree.  If the humidity
is above 99%, the vertical extent of cloud is estimated as the total number of adjacent levels with RH above the threshold. The
cloud fraction from the meteorological model is then used to scale how big fraction of the air-parcel  that is subjected to cloud
activation, in-cloud chemistry and eventually removal. Thus, if the model suggests 50% cloudiness, and if the air-parcel is at
an altitude where the RH is above the threshold, 50% of the aerosol is involved in the activation and the rest remain as is.
Likewise, if the model indicate precipitation (given as precipitation at ground level [mm hr$^{-1}$] representing the all precipitation
from the column above ground), only 50 % is affected by wet removal. The fraction removed per mm of precipitation is scaled
to the calculated liquid water path (LWP) [g m$^{-2}$] and column precipitation intensity.  If cloudiness is 100%, 100% of the
aerosol is subjected to activation and removal processes. This does however not automatically mean that 100% is removed. If
the cloud is thick and of high LWP, and precipitation rate is low, this will result in low scavenging rates in the air parcel. The
in-cloud scavenging scheme further assume warm clouds (i.e. without ice component).  In-cloud scavenging is subsequently
calculated from a modelled precipitation rate, assuming growth into precipitation sized droplets occurs via collision
coalescence between cloud droplets in the cloud The effect of ice components in clouds is neglected, which may alter the
scavenging efficiency and ultimately the wet removal rate. Clouds are prescribed, when relative humidity of the parcel is above
99%, a condition that initiates calculation of the cloud with a randomly chosen constant updraft between 0.1 and
1 m s$^{-1}$, with a normal distribution around 0.5 m s$^{-1}$, representing a typical stratocumulus cloud. Admittedly, being of crucial
importance for the lower size limit of activation, this approach has limitations, e.g. the crude assumption of the randomized
updrafts and it is not valid for convective precipitation or sub grid scaled clouds. Nevertheless, we argue that the range of
updrafts used reflect different cloud conditions ranging from typical low level stratus up to shallow convective clouds. The
box tracked along the trajectory is either below, in or above a cloud, if there is a cloud in the column where the box is situated.
If the box is in the cloud it is subject to calculation of droplet activation in-cloud scavenging, if it is below sub-cloud
scavenging. The in-cloud scavenging washes out only a fraction of the activated particles. This fraction is a function the cloud

water content (calculated from the liquid water path for the cloud) and ground level precipitation (which are parameters available in the meteorological fields used to calculate the trajectory). The below-cloud scavenging uses a parameterisation by (Laakso et al., 2003).

The difference in modelling wet deposition in CALM (a trajectory box model) and what is commonly used in a LPDM model is quite substantial. In CALM the removal dynamics is made to mimic what happens in a real cloud which traditionally has been too computationally heavy for inclusion in a LPDM. Traditionally in a LPDM, a scavenging coefficient that depends on meteorological parameters is used (Sportisse, 2007). The scavenging coefficients used for both in-cloud scavenging and below-cloud scavenging are often empirically determined. This inevitably creates a bias of the scavenging coefficients towards conditions typical for the chosen experiments and might thus not represent the variety of wet deposition that takes place in reality. For this reason alone it would be desirable to include more advanced wet deposition modelling in dispersion models.

To simulate something that a LPDM dispersion model could afford computationally a simplified version of the activation scheme was used with a fixed CCN activation size instead of one calculated from updrafts and humidity. This will henceforth be denoted *fixed activation size (FAS)* in the experiment description

| | Melpitz | Neo | Zeppelin |
|---|---|---|---|
| Year for the measurements | 2008 | 2012 | 2010 |
| Location, Latitude | 51.53 | 36.83 | 78.90 |
| Location, Longitude | 12.93 | 21.70 | 11.86 |
| Number of trajectories simulated Yearly coverage in % | 7995/8784 91% | 7769/8784 88% | 7178/8760 82% |
| Total number of simulations (5 experiments per trajectory) | 40060 | 39365 | 36030 |
| **Type of chemical background** | Central European background | Clean Remote background | Marine background |
| Ammonium bisulfate | 47 % | 45 % | 45 % |
| Insoluble organic compounds | 38 % | 42 % | 42 % |
| Primarily emitted species | 15 % | 13 % | 13 % |

Table 1: Stations for the simulations, data and initial chemical conditions.

**2.5 The different experiments**

For each of the trajectories we made five different experiments simulating single trajectories to analyse the impact of aerosol dynamics, summarized in Table 2. The experiments were designed in a way to, as transparently as possible, evaluate to what degree simulating advanced aerosol dynamics in LPDMs could influence the overall result. Identifying where and when detailed treatment could be beneficial is important in the context of emergency preparedness after a nuclear power plant accident:

  1. In the *first* experiment, we used the full setup with all aerosol dynamics including wet and dry deposition. This simulation represents the most detailed description of what a single trajectory of a LPDM dispersion model run would give as a result if it were to simulate all aerosol dynamic processes. Comparison with this experiment will show the effects of omitting certain processes. Abbreviation: ALL PROC.

2. In the *second* experiment, only dry deposition was active (no wet deposition and no other processes). This simulation represents the behaviour in the dry atmosphere without other aerosol dynamics. This is the simplest single trajectory setup representing a dispersion model mimicking either behaviour when there is no ongoing precipitation or a model omitting wet deposition totally. It will give us a sensitivity analysis the behaviour in the dry atmosphere if we were to neglect simulating all other processes. Abbreviation: ONLY DRY

3. The *third* experiment simulated dry deposition and clouds (including in-cloud and below-cloud scavenging) but no other aerosol dynamics. This is an analogue to the common approach in dispersion modelling where only wet and dry deposition are simulated. In this experiment however, the wet deposition scheme is more advanced than in many LPDMs using a scavenging coefficient. It includes below-cloud scavenging and in-cloud scavenging with CCN activation based on updrafts and available humidity (hencheforth denoted advanced wet deposition scheme or advanced cloud parameterization). This experiment is representing the behaviour of a LPDM with an advanced wet deposition scheme. Abbreviation: ONLY DEP

4. The *fourth* experiment was similar to the first experiment with all processes turned on but with at fixed CCN activation size (FAS) resulting in a simplified wet deposition scheme. This experiment represents the behaviour of a dispersion model with a simplified wet deposition scheme, but where the coagulation, condensation, emissions, chemical transformations and nucleation (henceforth denoted advanced aerosol dynamics) is included. Abbreviation: ALL PROC FAS.

5. The *fifth* experiment included dry deposition and wet deposition with the fixed activation size but no other processes. This experiment mimics the traditional LPDM without advanced aerosol dynamics and with a simplified activation scheme (which still has a more advanced wet deposition scheme than the traditional scavenging coefficient approach often used in dispersion models). Abbreviation: ONLY DEP FAS.

Note that including cloud interaction with a fixed activation size, FAS, as in experiment 4 and 5 is still a more advanced approach then the wet scavenging commonly used in LPDMs.

| Nr | Description, processes turned on | Abbreviation | Coagulation, condensation, emissions, nucleation | Fixed activation size | Drydep | Clouds and Wetdep |
|---|---|---|---|---|---|---|
| 1 | All processes | ALL PROC | YES | NO | YES | YES |
| 2 | Only Dry Deposition (no other processes) | ONLY DRY | NO | NO | YES | NO |
| 3 | Only Deposition (Dry dep and Clouds, including wet deposition) no other processes | ONLY DEP | NO | NO | YES | YES |
| 4 | All Processes are turned on, Fixed Activation Size (FAS) | ALL PROC FAS | YES | YES | YES | YES |
| 5 | Only Deposition (Dry dep and Clouds, including wet deposition) no other processes, Fixed Activation Size (FAS) | ONLY DEP FAS | NO | YES | YES | YES |

**Table 2: The different experiments**

## 3 Results

The outcome of the trajectory simulations described above is presented in this section. We have calculated "average trajectories" by combining the output of each individual time step counting from the start of the simulations. This was calculated into annual and monthly "average trajectories" with particle and caesium concentration throughout the 10 day

simulations for each of the sites. Normalized Becquerel represents [137]Cs where 1 denotes all released [137]Cs and 0.1 denotes 10 % of the released amount.

The evolution of the ambient PNSD of the aerosol and the size distribution of released [137]Cs (activity size distribution of the
[137]Cs attached on the particles) is shown in Fig. 4. This plot shows the annual mean average of the first experiment where all processes are turned on, ALL PROC. [137]Cs is represented as Becquerel normalized to the maximum value. The total particle number concentration and the total amount of [137]Cs attached on the particles are plotted as black lines in respective graph (right hand axes). The rows show Melpitz, Neo and Zeppelin respectively, dN/dlogDp to the left and [137]Cs (Bq) to the right. The development over the 10 day period is an effect of different routes of the individual trajectories, meteorological conditions,
new sources and of aerosol processes along the path. Even though each track has its individual characteristics and events, annual and monthly averages shows features representative for each site and period. The radioactive caesium is released one hour into each simulation and condensates on currently available particles, to be mixed with "clean" particles from new sources along the path.

For Melpitz, the average PNSD decreases after an initial peak one day into the simulation. The peak in the PNSD moves
slightly towards larger particles throughout the 10-day period. There is a very small but continuous increase in small particles (<10 nm) after the first day, throughout the simulation. The PNSD for Neo has a two modal initial distribution with many small particles (smaller than 10 nm). The total particle number concentration decreases after a small peak half a day into the simulation. After two days of simulation, the change in dN/dlogDp is barely visible even though the total particle number concentration decreases slightly. The PNSD in Zeppelin shows that it starts with a peak for particles smaller than 10 nm that
grows in both size and number the first 12 hours. After that the total particle number concentration decreases until late in day two where it starts growing slightly during the continuation of the simulations. The peak of the PNSD grows for Zeppelin throughout the whole simulation from around 10 nm to almost 100 nm.

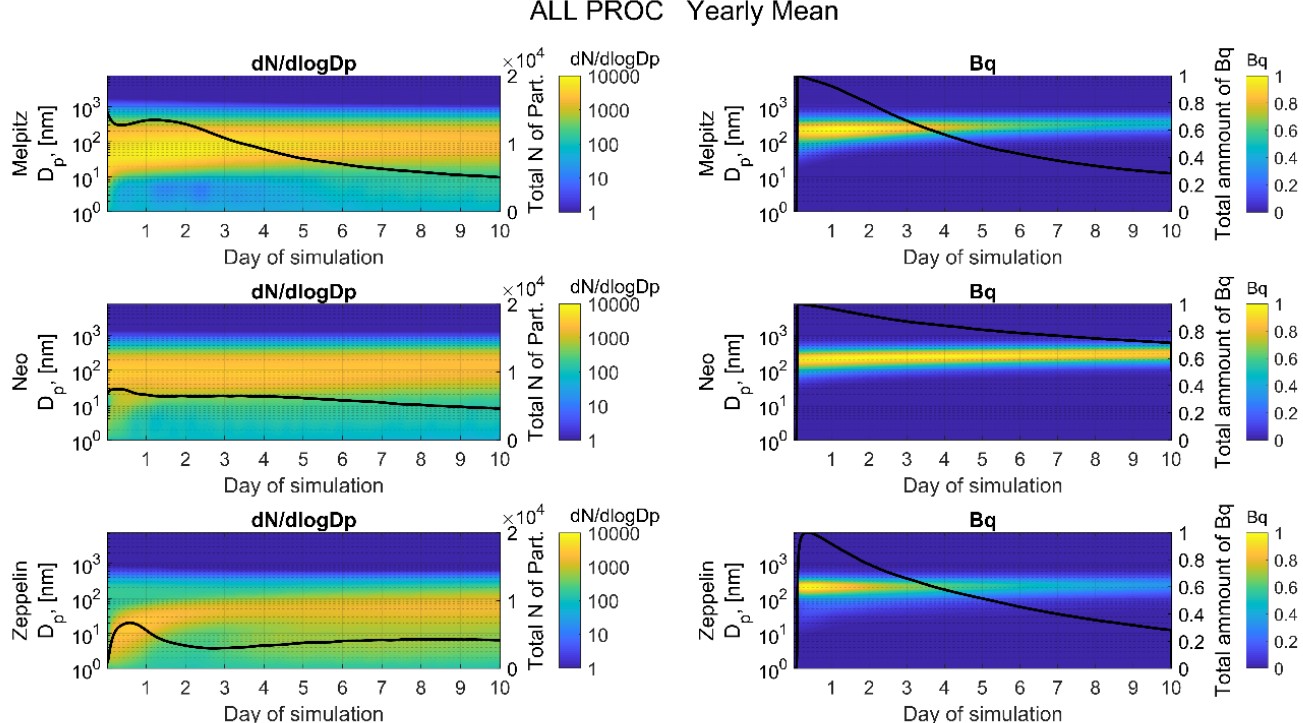

**Figure 4: Evolution of PNSD, dN/dlogDp (left column) and Becquerel size distribution (right column) for the experiment when all processes are turned on, ALL PROC, annual mean. Black lines are the total number of particles and the total amount of $^{137}$Cs in respective graph (right hand axes). Calculations from Melpitz, Neo and Zeppelin in the top, middle and bottom row respectively.**

The change in total $^{137}$Cs activity can be seen in Fig. 5 for the five different experiments (see Table 2), represented by normalized Becquerel (from 0 to 1). Each row represents one of the three stations and the columns the different experiments. The black line in each plot is the evolution of annual mean while monthly means are the lines with different colours. The December monthly mean has been omitted for the Zeppelin cases, since it had only 6 out of possible 744 simulations (0.9%), which was considered to be too few for statistical calculations.

The first column shows the experiment when all aerosol dynamic processes are turned on (ALL PROC). The different stations have different characteristics. The variability of monthly means at the end of the 10-day period is greatest for Melpitz (a spread of 52 %-points) and smallest for Zeppelin (a spread of 24 %-points) even though Zeppelin has a greater variability after 4 days (a spread of 50 %-points). Neo had a difference of 48 %-points at the end of the simulations. The summer months for Neo has a small decrease in caesium concentration. Neo has the smallest decrease of all the stations, down to 70% of the original amount compared to Zeppelin 28% and Melpitz, 27%. Melpitz has for all months a faster decrease in the first half of the simulation than in the second half. This is also true for the winter months in Zeppelin but not for summer and autumn.

The ONLY DRY experiment, second column, has only dry deposition turned on, i.e. no wet deposition and no other aerosol dynamic processes. The differences between the stations is due to the difference in the initial aerosol size distribution spectra

and its dry deposition along the trajectories. Dry deposition only occurs when the box is in the mixing layer so the height variation of the box also plays a role. The variation in monthly means after 10 days is greatest for Zeppelin (a spread of 27 %-points), smallest for Neo (a spread of 10 %-points) and Melpitz (a spread of 16 %-points). The total decrease for the annual mean was for Zeppelin down to 80%, Melpitz 75% and Neo had the smallest decrease to 92% of the released amount.

ONLY DEP, the experiment in the middle column, includes both dry deposition and advanced cloud parameterization with wet deposition, but no other aerosol processes. The difference between this experiment and the previous one comes from the effect of wet deposition. This result is similar to the first experiment, ALL PROC, where all processes are taken into account. Both share the same advanced cloud-processing scheme.

ALL PROC FAS, the fourth column, simulates the impact if we include all aerosol dynamic processes but have a fixed

activation size for the cloud droplets. This experiment differs from ONLY DEP both in monthly spread and in total decrease of caesium. The spread is for Melpitz and Zeppelin 32 %-points and for Neo 29 %-points. The total decrease is smaller than for previous experiments with air/cloud interaction and wet deposition (ALL PROC and ONLY DEP) and higher than then experiment with only dry deposition (ONLY DEP). The Melpitz $^{137}$Cs concentration is down to 45 %, Neo only to 80 % and Zeppelin to 56 % of the initial amount.

The last experiment (the right column) simulated only dry deposition and wet deposition though clouds with a fixed activation size, ONLY DEP FAS (no other aerosol dynamic processes). The output of this experiment has similarities with, ALL PROC FAS and they share the same simplified cloud activation scheme. For Melpitz the monthly spread in $^{137}$Cs concentration was 33 %-points after 10 days, close to ALL PROC FAS. For Neo the spread was slightly smaller than ALL PROC FAS, 17 %-points and for Zeppelin 27 %-points. The total decrease in $^{137}$Cs concentration was for Melpitz down to 57 %, for Neo to 85

430  % and for Zeppelin to 63 % of the released amount.

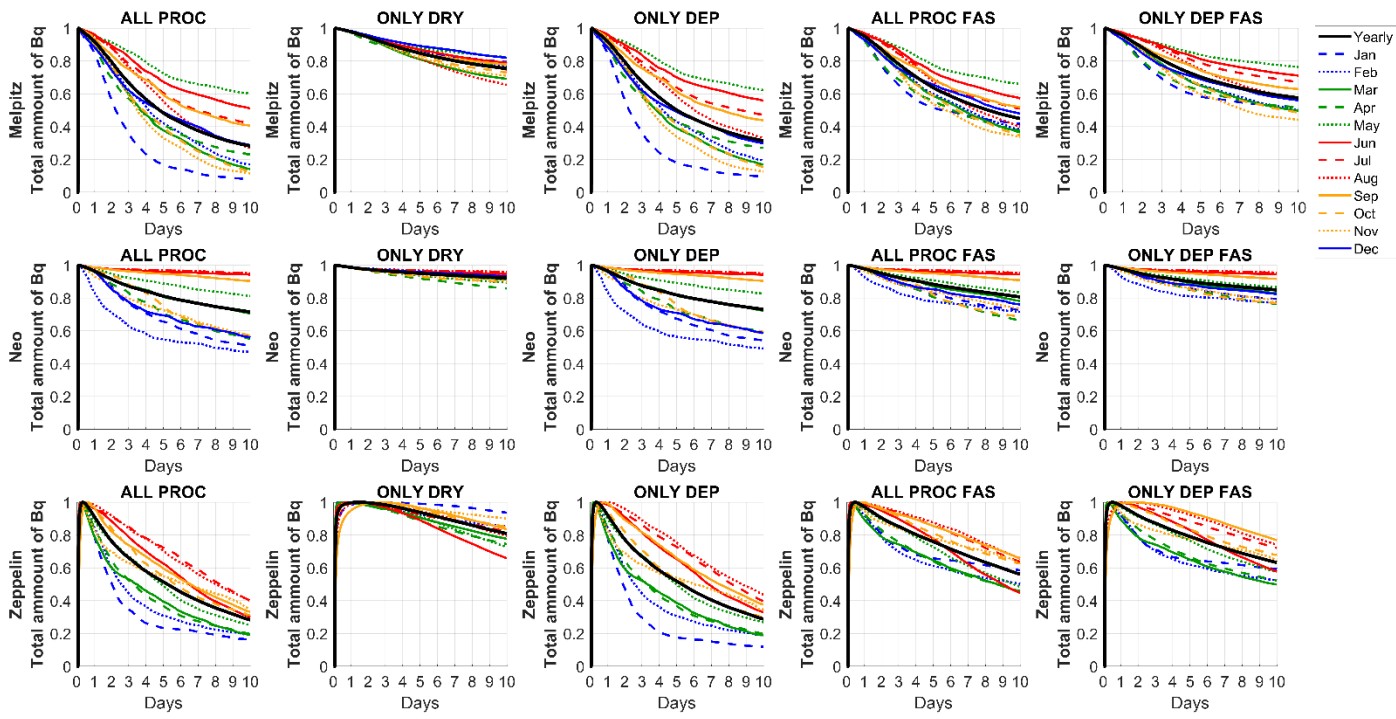

**Figure 5: Annual (black) and monthly means (blue for winter, Dec, Jan and Feb, green for spring, Mar, Apr and May, red for summer, Jun, Jul and Aug and orange for autumn, Sep, Oct, Nov). Development of caesium concentration as normalized Becquerel for the five different experiments. Each row represents the different stations (from the top, Melpitz, Neo and Zeppelin) and the columns the different experiments, ALL PROC, ONLY DRY, ONLY DEP, ALL PROC FAS and ONLY DEP FAS.**

Wet deposition is the by far the most efficient removal process for accumulation mode particles due to slow diffusion and terminal velocity. For reference, the total accumulated precipitation for each trajectory is shown in Fig. 6. Grey curve is the individual trajectories and black curve is a moving 3-day mean. The periods without size distribution measurements at the stations, and therefore no simulations, are visualised as red periods.

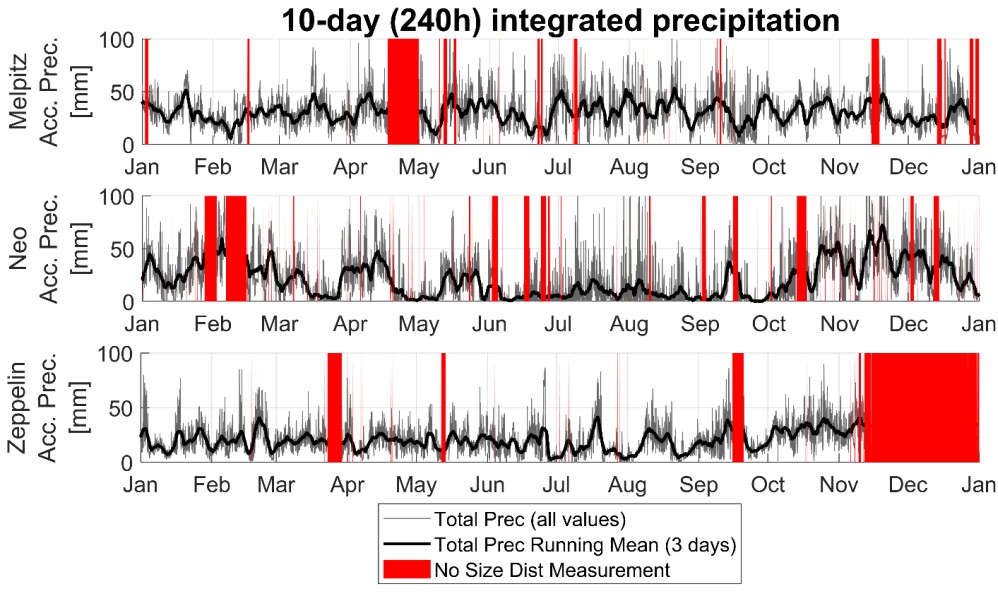

**Figure 6: Total accumulated precipitation for each 10-day trajectory for the three stations, grey line. The black line is a moving 3-day mean and red areas are periods with no input data (no size distribution measurement, hence no simulations). The y-axis is set to 0-100 mm even though there are a few peaks with higher values.**

The coupling between caesium concentration at the end of the 10-day simulations and the accumulated precipitation is illustrated in Fig. 7 The trajectories are binned with respect to the accumulated precipitation i.e. all trajectories with 0-10 mm accumulated precipitation is in the first, 10-20 in the second and so forth. Mean values for all the trajectories in the respective bin are shown, one line for each experiment. Blue bars are number of trajectories in each bin (the right hand y-axes). Note that there were too few trajectories in bins with high amount of precipitation for statistics and bins with fewer than 30 trajectories

have been filtered out. The maximum amount of accumulated 10-day precipitation is 100 mm for Melpitz, 150 mm for Neo and 99 mm for Zeppelin and the lines ends where there are no more cases. The ONLY DRY simulation (red line) is not affected by precipitation and the variations due to the amount of precipitation is more coincidental but is visualised for reference.

For all stations, it is clear that the two experiments with advanced cloud simulations, ALL PROC (blue line) and ONLY DEP (yellow line) have the biggest reduction in caesium. They are also very similar to each other even though ONLY DEP has

455 mostly slightly less remaining caesium. The two simulations with fixed activation size, ALL PROC FAS and ONLY DEP FAS, have a substantially higher amount of caesium left compared to ALL PROC and ONLY DEP.

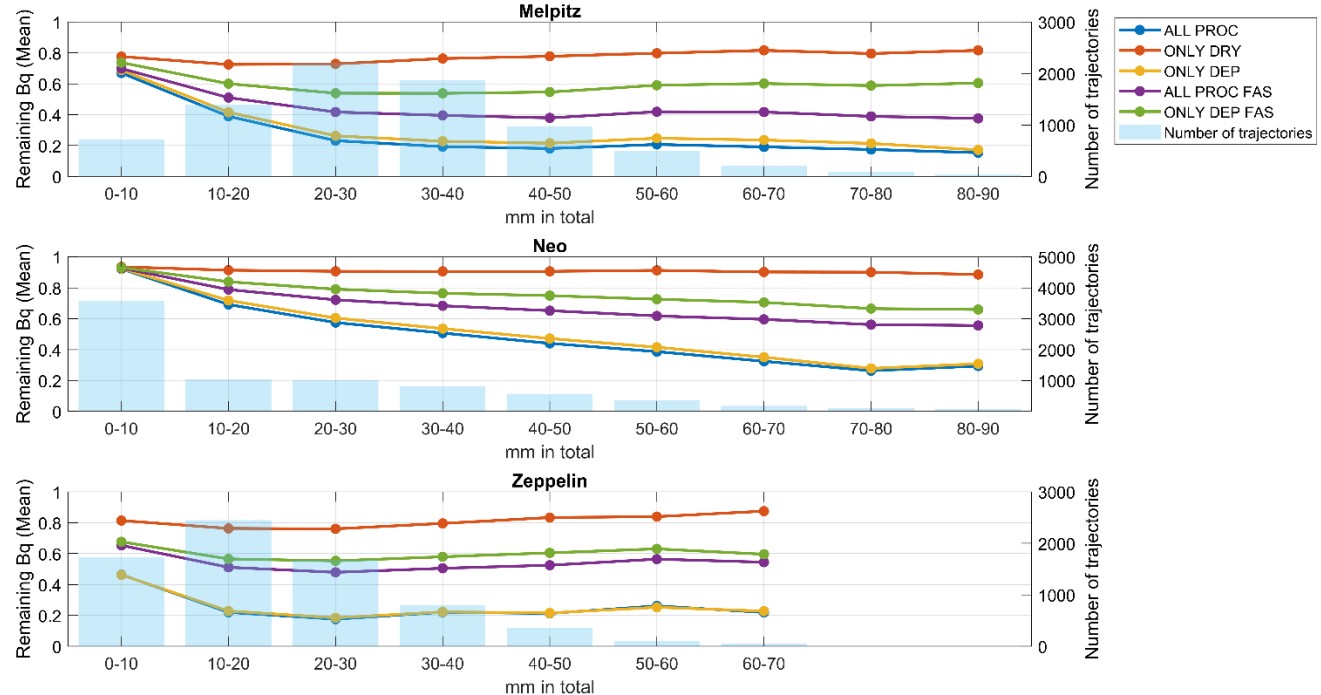

**Figure 7: Total accumulated precipitation and caesium concentration at the end of the 10-day trajectories. The trajectories are binned with respect to its total accumulated precipitation (x-axis). The mean final caesium concentration for all trajectories in each bin is plotted (left hand y-axis). The unit is normalized Becquerel for each of the experiments. The number of trajectories in each bin is shown in blue bars (right hand y-axes).**

The experiments with advanced cloud parameterization (ALL PROC and ONLY DEP), have for Melpitz and Zeppelin an increasing caesium removal with more precipitation for the first bins (four bins for Melpitz 0-40 mm and three bins for Zeppelin, 0-30 mm). Then the removal rate flattens out for higher precipitation amounts and there is no clear change in the remaining caesium concentration. Note that there are very few trajectories in the bins with high amount of precipitation and changes here reflects more the behaviour of individual trajectories then a statistical average. The initial decrease of remaining caesium with higher precipitation amounts is also true for ALL PROC FAS and ONLY DEP FAS but the decrease is not as strong as for ALL PROC and ONLY DEP. In Neo the final caesium concentration decreases between 0-80 mm of accumulated precipitation. For higher precipitation values, there is more irregular variations for Neo, that especially for the higher bins reflect individual trajectories more than a statistical average. This suggest that the fraction of aerosols available to wet removal decrease with time, and that cloud formation mainly make use of particles without $^{137}$Cs formed during the simulations and thus does not affect the total $^{137}$Cs concentration to a large extent. Thus, when the initial precipitation events have removed the bulk of Cs-containing particles, the remaining $^{137}$Cs-containing particles are few and cloud droplet activation takes place on newly formed particles without $^{137}$Cs. The remaining caesium for Melpitz reaches 19 % of the released amount for ALL PROC,

23 % for ONLY DEP, 40 %for ALL PROC FAS and 54 % for ONLY DEP FAS. For Zeppelin the remaining caesium decrease until 20-30 mm of precipitation. Higher levels of precipitation does not bring down the concentration much more (se comment about high precipitation levels above). Remaining caesium reaches 18 % for ALL PROC and ONLY DEP, 48% for ALL PROC FAS and 55 % for ONLY DEP FAC. For Neo the decrease of remaining Becquerel does not flatten out as early, instead

it continue to decrease with accumulated precipitation up to 70-80 mm. The air concentration after the 10-day period is then 27 % of the initial amount for both ALL PROC and ONLY DEP, 56 % for ALL PROC FAS and 66 % for ONLY DEP FAS. Figure 7 shows the seasonal variation for the remaining caesium concentration after the 10-day simulations (monthly means in normalized Becquerel). The largest reduction of caesium air concentration can be seen for all stations in the experiments ALL PROC and ONLY DEP, the experiments with advanced cloud parameterization. For these two experiments, there is a

clear seasonal variation with higher remaining concentrations in the summer for Melpitz and Neo. Melpitz has the highest remaining concentration in May, 60 %, and the lowest in January, 8%. Neo has the highest values, 95 %, in the three months of Jun-Aug and the lowest value 47 % in February. The seasonal variability is not as strong for Zeppelin as it is for Melpitz and Neo.

The individual order between the different experiments are the same throughout the year for all three stations (except for the

minor differences between ALL PROC and ONLY DEP in Zeppelin). ONLY DRY has the most remaining caesium (smallest reduction), then comes ONLY DEP FAS, ALL PROC FAS, and finally ONLY DEP and ALL PROC in the written order. The last two experiments are very similar in most cases.

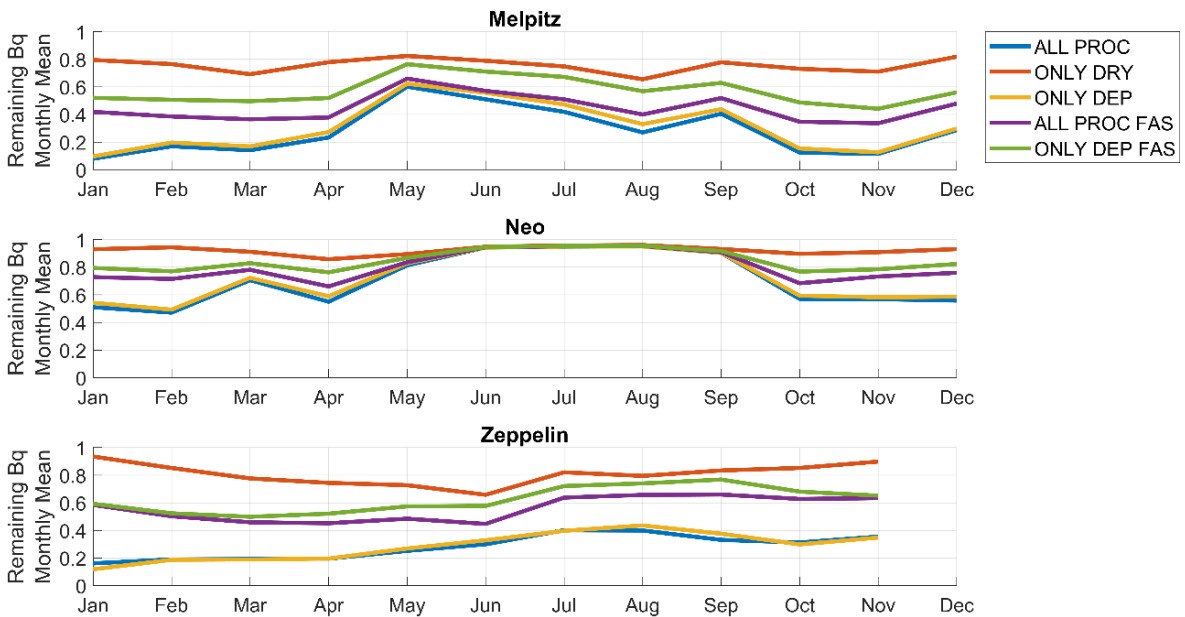

**Figure 8: Remaining caesium air concentration after the 10 day trajectories. Seaonal variation of monthly means for the different**
**sites (Melpitz, top, Neo, middle and Zeppelin bottom).**

The previous figures have described the monthly and yearly mean averages. To better account for variability within the averaged time periods, we have analysed the behaviour of individual trajectories, thereby focusing on differences as percentile values for the different experiments. To analyse the impact of the full aerosol dynamic parameterisation these trajectory-differences has been visualised in Fig. 9. The trajectory-difference is defined as the difference of caesium air concentration at
each time step in the 10-day simulation for two identical trajectories with different model physics (different experiments). It is shown as the difference in %-points between the two experiments.

The trajectory-difference between the experiments with the full model, ALL PROC, and the experiment with only dry deposition and cloud parameterisation, ONLY DEP, is shown in the left column (ALL PROC minus ONLY DEP). This trajectory-difference represents the impact that advanced aerosol dynamics has, when added to a model using dry deposition
and advanced cloud parameterization with wet deposition. The added processes include coagulation, condensational growth, nucleation and interaction with the background aerosol involving new sources then creates the deviation from the 0-line in these plots.. The trajectory-difference between ALL PROC and ONLY DEP FAS is shown in the right column (ALL PROC minus ONLY DEP FAS). This shows the difference between a model with dry deposition and simplified cloud parameterisation mimicking a LPDM simulation, and a full-scale aerosol dynamic model (including advanced cloud
parameterisation and advanced aerosol dynamics). The percentiles in Fig. 9 shows that there are outliers that does not follow the 0-line. Graphs on the 0-line indicates that there is no difference between the different experiments. The line for the $X^{th}$ percentile shows the magnitude of the difference between the two experiments in X% of the trajectories (e.g. the $5^{th}$ percentile represents the difference for 5% of the simulations).

After the 10 day period the $5^{th}$ percentile is -11% for Melpitz, -9% for Neo and -7 % for Zeppelin for ALL PROC – ONLY
DEP. The median ($50^{th}$ percentile) is slightly lower than the 0-line in Melpitz, -2 % after 10 days. In 75% of the cases ALL PROC is smaller than ONLY DEP (the $75^{th}$ percentile is slightly negative). It means that in 75% of the cases including advanced aerosol dynamics makes the removal of caesium in the atmosphere more efficient. This is not true for Neo and Zeppelin where the median ($50^{th}$ percentile) follows the 0-line. For all sites, there is a positive peak in the trajectory-difference for the $75^{th}$, $90^{th}$ and $95^{th}$ percentile in the beginning of the simulations, peaking just after the release. It originates from the
release of the caesium gas and the condensation process that differs slightly between the experiments. This is most pronounced in the Zeppelin case where the surface area of ambient particles available to condensate on, is smaller due to fewer particles in Zeppelin. Since there are both negative and positive percentiles, including advanced aerosol dynamics can both increase and decrease the estimated air concentration of caesium. The magnitude of the trajectory-difference is however larger when the ALL PROC simulations has lower concentrations than ONLY DEP.

The trajectory-difference between a wet deposition scheme mimicking a LPDM simulation, ONLY DEP FAS, and simulations with full aerosol dynamics, ALL PROC, can be seen in the right column in Fig. 9. The median for both Melpitz and Zeppelin is around -30 %-points after the 10-day simulation (30 %-points difference if using a simplified cloud interaction parameterisation compared to full aerodynamic simulations). The median in Neo is following the 0-line. For 5 % of the simulations there is around -60 %-points difference already after 5 days for all three stations. The $10^{th}$ percentile for Melpitz

is after 5 days -50 %-points, Neo -43 % points and for Zeppelin -54 %-points. In this comparison all cases (except for Zeppelin during the initial caesium condensation period) all percentile values are negative. The simulations mimicking a LPDM over predict the air concentration of released caesium. This will in turn under predict the deposition field.

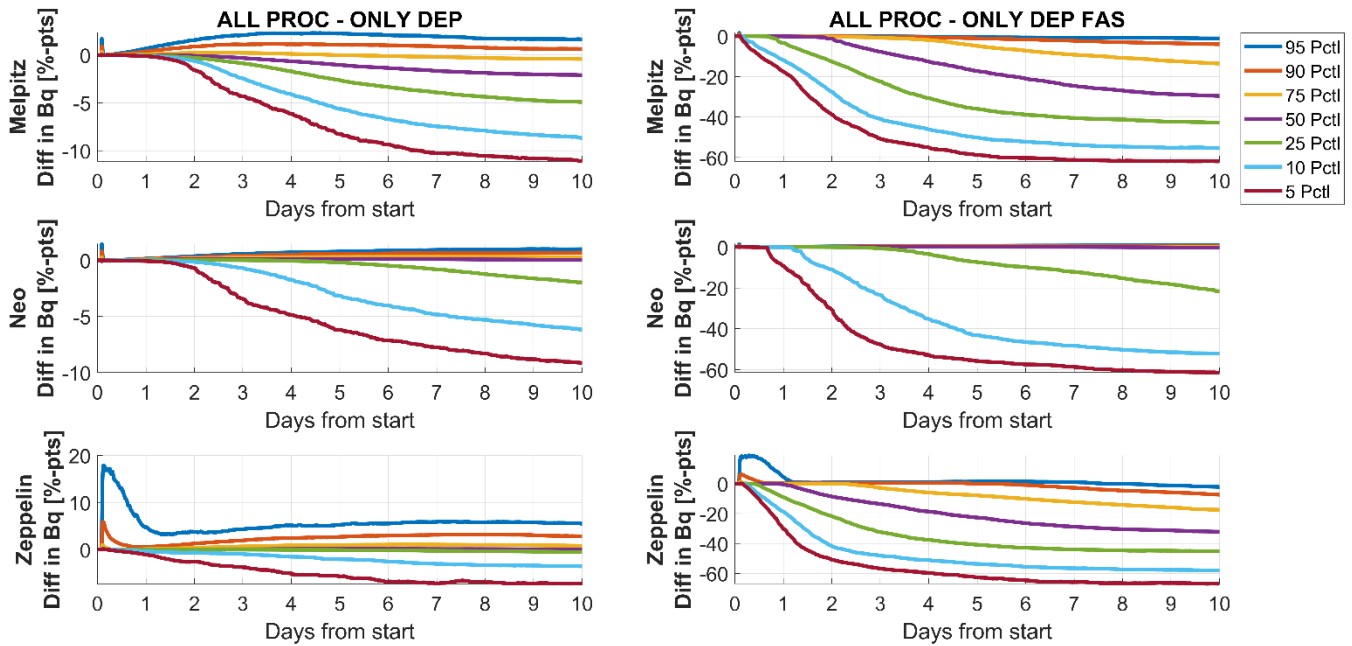

Figure 9: Difference in %-points for the total caesium concentration for two different experiments. Percentile values over the 10-
day period. ALL PROC minus ONLY DEP, left column and ALL PROC minus ONLY DEP FAS, right column. The top row is for Melpitz, middle row for Neo and the bottom row for Zeppelin. Negative values represent when ALL PROC has lower concentration than the compared experiment (ONLY DEP, left and ONLY DEP FAS, right) i.e. when advanced aerosol dynamics gives lower air concentrations. Note that the scales of the y-axes vary between the sub plots.

## 4 Discussion

Considering the results in the previous section and keeping in mind the application for radiological emergency preparedness we would like to highlight a few observations.

There are many different conditions and processes playing a role in the outcome of the simulated scenarios. The initial size distribution, Fig. 3 represents a key aspect in the setup of this study. The variability of the initial PNSD reflect both local and regional properties of the ambient aerosol and their relation to the sources, sinks and processes along the path it has taken to

arrive at the site in question. During transport in the atmosphere, the interaction between condensation, coagulation and cloud processing (excluding precipitation) transforms an aerosol into the accumulation mode. Large particles in the coarse mode are rather rapidly removed due to gravitational settling, and particles in the smaller end of the size spectrum are rapidly grown into larger sizes through condensation, or due to their small size removed either by dry deposition or by coagulation with larger

sized particles. Once in the accumulation mode, the only way of effectively removing the particles is through wet deposition in general and in-cloud scavenging in particular. As the lifetime for nuclei mode particles and coarse mode particles is short, long range transport of Bq is largely controlled by on the one hand how rapidly Bq get partitioned into the accumulation mode, both directly via gas-particle-conversion or indirectly by growth of smaller particles containing radioactive tracers. Once in the accumulation mode, the lifetime will largely be controlled by intensity and frequency of precipitation. The amount of time needed to partition the $^{137}$Cs into the accumulation mode depends on how much of the aerosol that already is there when the radioactive release starts. The characteristics of the initial aerosol PNSD also determines the effect that the different aerosol processes will have. Further, this will also connect to how the radionuclides initially are described. In our study we have assumed that the nuclides are emitted as a low volatile vapour. Other assumptions could be emission of a pure combustion aerosol that dynamically will interact with the background aerosol. The nature of emission scenario will of course therefor also affect the outcome of the study. The source fields that the trajectory box model travels through determine, together with the sinks, what new non-radioactive particles the radioactive aerosol will interact with in the simulated box. This occurs in this model setup when the box is within the mixing layer (depending on the trajectory path) which will make the PNSD differ from the $^{137}$Cs activity size distribution. We have not studied the complexity near the source within the first day where aerosol dynamic processes might have a different impact, especially when it comes to releases in highly polluted environment. To address the situation near the source other type of dispersions models are more suited for purpose than LPDMs which is not the scope of this study. The focus here has been on the result during and at the end of the 10-day simulation to understand the role of aerosol dynamics on this time scale. The *meteorology* clearly plays an important role for the outcome of each trajectory. It determines the path of the trajectory, horizontally as well as vertically. The meteorology also determines the conditions for clouds and therefore the air/cloud interactions and wet deposition, as well as the conditions for processes dependant on humidity, temperature and pressure. Since wet deposition is the most effective sink of the accumulation mode, the cases where there are clouds and precipitation have very different dynamics than the ones without clouds. The *dynamical processes*: condensational growth, coagulation, dry deposition, nucleation and chemical interactions takes part in transforming the aerosol. They determine how the initial radioactive aerosol together with new sources are transformed into the accumulation mode. There is a difference in chemical composition for the particles in the different trajectories. This is important for droplet formation and wet deposition that depends on the hygroscopicity of the particles. However, the focus in this study have been less on the dependence of chemical composition and more on the resulting PNSD and activity size distribution of caesium. The change of the PNSD for the released radioactive caesium, attached on the surrounding particles is visible in the annual averages in Fig. 4 and there are small but visible differences between the different sites. The decrease in total caesium concentration is much stronger in Melpitz and Zeppelin than in Neo. There are not so strong changes in the $^{137}$Cs activity size distributions (relative change between small and large particles) in the annual averages. What can be seen is the just mentioned decrease for Melpitz and Zeppelin and an ever so slight increase in the sizes of the caesium particles for Neo. Part of an explanation for this is that that the initial surface area size distribution, Fig. 3, is often already located close to the accumulation mode around 0.1-1 µm to where the aerosol processes strive to transform the aerosol. Wet deposition, the strongest sink for

the aerosol in the accumulation mode is directly correlated with the precipitation. It explains the small change for the amount of caesium for Neo, since 46 % of the trajectories have very little precipitation, 0-10 mm for the 10 day simulation, Fig. 6.

Many of these cases are in the period June to mid-October, compare with total precipitation in Fig. 6, and it makes an impact on the annual average. It can also be seen in Fig. 8 that the reduction of caesium is small in the period June-September for Neo.

The impact of a good description of cloud interactions and wet deposition is visible in Fig. 7. The experiments with full cloud parameterization (ALL PROC and ONLY DEP) have a substantially more efficient removal of caesium than the experiments

with fixed activation size (ALL PROC FAS and ONLY DEP FAS. The findings suggest that the impact of aerosol dynamics is of lesser importance for the average dispersion and deposition patterns given that the wet deposition is handled sufficiently accurate. The result for ALL PROC and ONLY DEP are in this case very similar for all sites. If, for some reason, it is not possible to parameterize the cloud droplet activation well, then there is a benefit for including advanced aerosol dynamics (compare ONLY DEP FAS and ALL PROC FAS), since it decreases the air concentration further towards the experiment

simulating all aerosol dynamic processes (ALL PROC), Fig. 8. This is visible for all stations even though the effect is biggest for Melpitz (see the difference between ALL PROC FAS and ONLY DEP FAS).

The scavenging coefficient, a commonly used method both for in-cloud scavenging and below-cloud scavenging in a LPDM, is often derived from experiments. This inevitably creates a very close relationship between the model and the chosen experiments. This might not represent the variety of wet deposition that takes place in reality The fixed activation size method,

FAS, describes the creation of cloud droplets that might lead to wet scavenging of the aerosols which makes it more physically credible compared to a scavenging coefficient. The FAS method includes where in the column above and below the box cloud formation and precipitation occurs whereas the scavenging coefficient only correlates ground level precipitation to the wet deposition with no height dependency. The coupling between activated particles and remaining particles for future activation is however neglected in the FAS parameterization. The activation size varies depending both on the abundance of water vapour

but also on the available particles. If there is still an environment for making new cloud droplets after a droplet making event and all particles larger than the previous activation size are gone then the activation size becomes smaller leading to even smaller particles being activated. It makes this scheme sensitive for the choice of fixed activation size and calculating this from current meteorological conditions and the current PNSD would be preferable. In the future, it would be interesting to have access to more cloud parameters from the numerical weather prediction models for better simulations of the air/cloud

interaction and wet deposition

The result of all the simulations when it comes to the *mean values* shows that including advanced aerosol dynamics does not always make a huge impact on the resulting caesium concentrations. It can be seen in Fig. 9 the median (50th percentile) difference between with and without aerosol dynamics in the case with an advanced cloud parameterisation scheme (left column) is either close to zero for Neo and Zeppelin or 2 %-points for Melpitz (at the end of the 10 day simulation). It implies

that for statistical simulations, LPDMs can perform well without advanced aerosol dynamics as long as they model cloud physics closely. However, for decision support in emergency preparedness, the mean values are irrelevant; instead, the single

realisation at hand has to be simulated (for the current location and weather situation in question). In the 5th percentile there is a 60 %-points difference if aerosol processes together with an advanced cloud scheme are modelled (compared to using the simplified cloud interaction scheme, FAS, right column Fig. 9). Also in the 25th percentile, there is a large difference approx.

45 %-points in the end of the simulation for Melpitz and Zeppelin, 20 %-points for Neo. In a situation of radioactive hazard management, this difference could change a decision of whether to evacuate or not evacuate a certain area or how to deal with aspects of the food industry, crops and cattle. Taking severe actions brings also higher costs and complications for society and industry. The focus in this study is airborne concentration of the radioactive material since that is the outcome of this model setup, while use of a dispersion model also calculates ground contamination (deposition fields). The conclusions made from

airborne radioactivity concentration can be transferred to ground contamination since they are directly linked. When the airborne concentration of radioactive material is reduced by wet deposition it will create hotspots of deposited material on the ground. The location of these hotspots are very sensitive to each precipitation event, which may not be obvious when only analysing the air concentrations. Identifying these hotspots are still very important regarding migitating actions. This could be a topic for a future study since ground contamination is not a part of this study.

It can also be worth noting that including advanced aerosol dynamics, ALL PROC, compared to ONLY DEP (both using advanced air/cloud parameterisation), can lead to both higher and lower air concentration, (cf. Fig. 9, left frames). The percentiles are distributed on both sides of the 0-line. For Melpitz the 90th and 95th percentile is positive (higher concentration in ALL PROC than in ONLY DEP) and the rest are negative. For Neo the 50th percentile and higher are positive but quite close to zero and the rest are negative with the 5th percentile on -9 %-points after 10 days. The percentiles for Zeppelin on the

other hand has a quite equal distribution on both sides of the 0-line with a maximum of 6 %-points for the 95th percentile and -7 %-points for the 5th percentile. It shows that advanced aerosol dynamics can either increase and reduce the air concentration depending on the site location and current weather parameters (while using advanced air/cloud parameterisation).

As an illustration, we consider one realisation (belonging to the 5th percentile) in Fig. 10. The top row shows PNSD and the bottom row show the caesium distribution over time for the three different experiments, ALL PROC, ONLY DEP and ONLY

DEP FAS, the same experiments used in the trajectory-differences in Fig. 9. The black line represents the total number concentration and caesium concentration in resepective plots. The trajectory enters a precipitation region after 3.5 days which can be seen in the ALL PROC simulation (left) as big particles are reduced (>100 nm) and all the caesium located on those particles disappear quite instantly. In the ONLY DEP simulation (middle) the big particles also disappear but not all caesium. The caesium distribution in this case is the same as direclty after the release and the appearance remains throughout the

simulation (no advance aerosol dynamic processes are simulated). Therefore, a part of the caesium still remains in the air thoughout the 10 days even though the bigger particles have deposited.

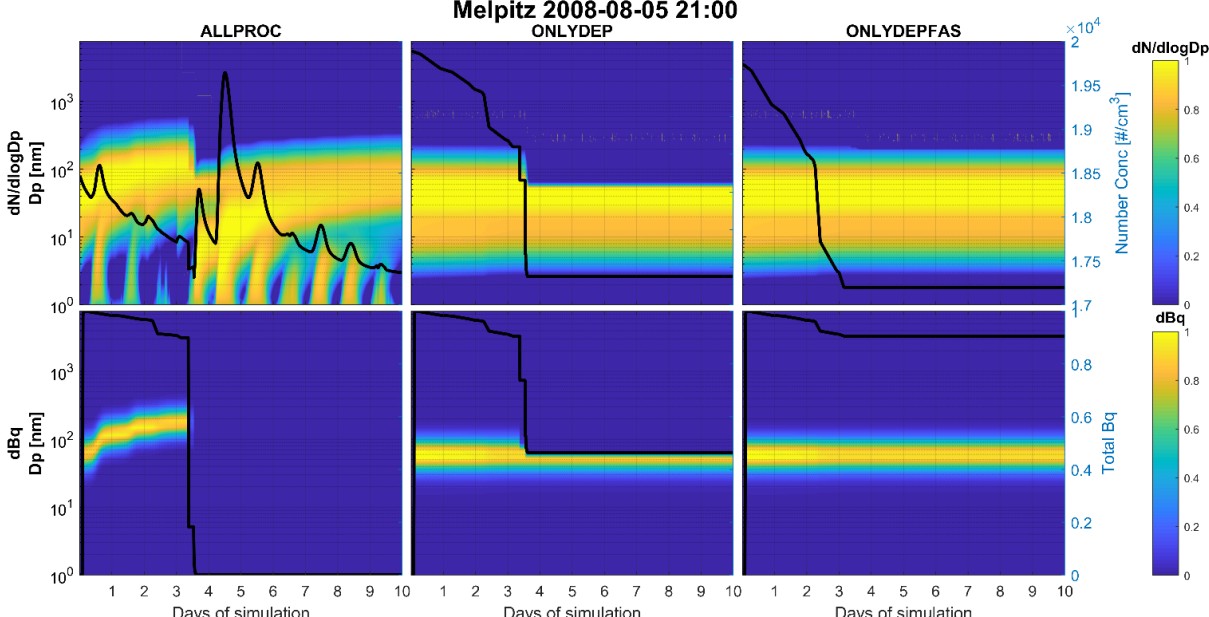

**Figure 10: Result from one trajectory, Melpitz 2008-08-05 21:00. Evolution of size distribution spectra for number concentration, dN/dlogDp (normalized) top row, and caesium concentration, dBq (normalized) bottom row. Three different experiments, ALL PROC (left column), ONLY DEP (middle column) and ONLY DEP FAS (right column). The black lines represent total number concentration and total Becquerel concentration respectively (axes to the right).**

The activation size for ALL PROC and ONLY DEP is calculated from updrafts and humidity and this size is clearly visible in the caesium concentration plot in Fig. 10, middle column, since all particles above 60 nm are removed after about 3 days. In the ONLY DEP FAS simulation (right column Fig. 10) almost all caesium is located on particles smaller than the fixed activation size which makes the washout of caesium much less effective. After the precipitation event the remaining caesium is in the ALL PROC simulation 0 %, ONLY DEP 46 % and ONLY DEP FAS 90 % of the initial amount. These levels are reached directly after the precipitation period after about 3.5 days and stays throughout the rest of the simulations. This shows that advanced aerosol dynamics in ALL PROC transform the particles, with aid of coagulation and condensational growth, bearing caesium into a size range where wet deposition via activation is predominant. Without this growth, the size of the Cs-bearing particles will not reach the size required for activation into cloud droplets. The result will be less efficient removal.

Comparing seasonal variations Neo stands out (Fig. 8). Neo has very low loss of Becquerel concentration in the air during the summer. This is due to the low amount of precipitation and clouds during the summer months for the Neo trajectories (see Fig. 6). Seasonal variations can also be seen in the Melpitz simulation especially in the experiment when all processes are turned on (cf. Fig. 8). This originates most likely from the initial caesium size distribution. The surface area size distribution of the measured ambient particles for the different stations can be seen in Fig. 11. Mean distributions are shown for the periods Jan-Apr, May-Sep and Sep-Dec. The periods are chosen from Fig. 8 to emphasise the differences of the characteristic seasons. The summer period for Melpitz, May-Sep, has the smallest surface area size distribution. This leads to less washout with the

advanced air/cloud parameterisation schemes, since the number of particles that are activated into cloud droplets are fewer, hence the seasonal variation in Fig. 8.


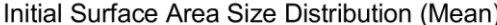
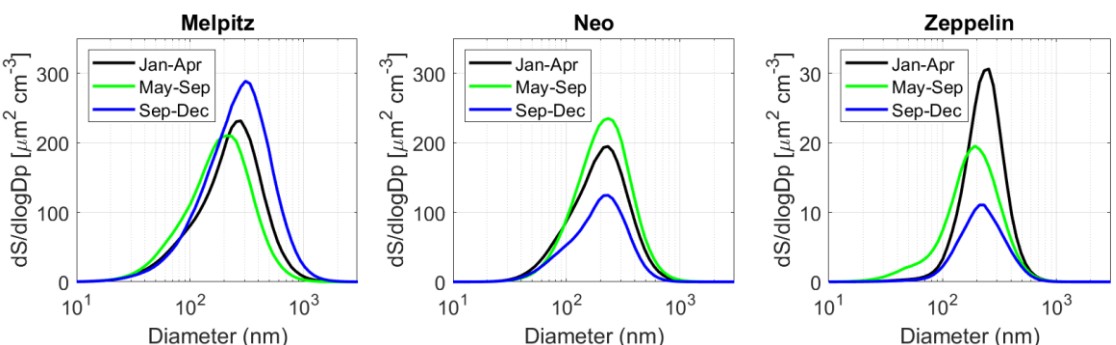

**Figure 11: Surface area size distribution for measured ambient aerosol used to initiate each trajectory. Mean distributions for spring (Jan-Apr), summer (May-Sep) and fall (Sep-Dec) for Melpitz, Neo and Zeppelin. The division into these time periods is taken from to emphasise differences between different periods. Note that the Y-axes have different scales for the different sites.**

The effect of turning on and off processes is stronger (larger spread between the different experiments) for Melpitz and Zeppelin than for Neo in Fig. 8. Especially in the beginning and the end of the year. For Zeppelin the seasonal variation is stronger after 5 days of simulations then after 10 days see Fig.4. In ALL PROC for Zeppelin the spread of monthly curves are much bigger around 5 days then at the end of the 10-day simulations, where they converge. It is clear in Fig. 11 that the seasonal variation in peak size of the particles is almost none in Neo but vary for both Melpitz and Zeppelin with the smallest

diameters in summer. Melpitz has the most accumulated precipitation of all stations and Zeppelin the least. However, the effect of the simple cloud interaction scheme (fixed activation size) is strong for both Melpitz and Zeppelin even though the stations differ in total accumulated precipitation. The total number of particles in the air at Zeppelin vary over the year, Fig. 3, but compared to the other stations the total number concentration is generally low. When there are circumstances for droplet activation though, even smaller particles gets activated since the water vapour uses available particles for condensation. This

leads to the biggest washout after the 10 days if all months are considered, Fig. 4, even though Zeppelin has the lowest amount of total precipitation of the three stations.

It can be argued that once a particle enters the accumulation mode size range, its fate is largely controlled by the frequency and intensity of precipitation. Once this size has been reached, the particle will be comparably inert towards changes induced by condensation and coagulation, and will retain its characteristics over long timescales. When the accumulation mode size

range is reached more simplified physical parameterisation might be sufficient for the continuation of the simulation allowing for first order treatment by either in-cloud or below-cloud scavenging. This would substantially limit computational costs compared to the concept of including all physiochemical aerosol dynamic modelling throughout the whole simulation.

**5 Conclusions**

To analyse the impact of including more advanced aerosol dynamics in LPDM simulations for radioactive releases we have
simulated single trajectories with a Lagrangian moving box model. For three different sites, a year of hourly ambient PNSD
measurements initiated the model. To emulate a nuclear power plant failure we released $^{137}$Cs into the atmosphere, which then
condensated on the ambient aerosol. The change of the PNSD and the radioactive activity size distribution were tracked all
through a simulation time of 10-day simulation. Five different experiments were made for each of the over 22000 trajectories.
The experiments represented different setups of the simulated processes. The simulated processes included coagulation,
condensational growth, nucleation, additional sources of non radioactive particles along the path, chemical interactions, dry
and wet deposition including aerosol/cloud interactions.

Comparing the mean values for the experiments simulating only dry and wet deposition including advanced cloud interactions
and the full aerosol dynamic simulations show small differences. For single events however the differences can be larger. For
long term statistical dispersion modelling having a good particle/cloud interaction scheme can therefore be sufficient. In a
radioactive emergency situation, which is the topic of this study, single events can deviate from the statistical result and more
advanced parameterization might be necessary to adequately capture air concentration and deposition of radioactive material.
Precipitation brings the accumulation mode particles to the ground, which emphasise the importance of a good aerosol/cloud
interaction parameterization scheme with wet deposition. We conclude that a good aerosol/cloud interaction scheme is the
most important of the aerosol dynamic processes. If this is used together with dry deposition, the impact of including remaining
aerosol processes can be described by the values of the concentration at the end of the 10 day simulations. We studied the
magnitude of the difference between including or excluding the remaining aerosol processes for each trajectory in a statistical
sense by considering percentiles. For 5% of the simulations (5th percentile) there is 10%-points higher air concentration for
all stations when only using the advanced aerosol/cloud interaction scheme as well as wet and dry deposition. In these cases,
the ambient dry aerosol processes increase the wet deposition. The 95[th] percentiles for Zeppelin and Melpitz are positive which
means that when all aerosol processes are included the radioactive particles are less prone to be activated for wet removal.

An aerosol/cloud interaction scheme with a fixed activation size is even more sensitive and generates a 60%-point difference
for the 5[th] percentile of the air concentration of $^{137}$Cs after 10 days. That is, the air concentration is 60%-points higher for the
fixed activation size run compared to using the full aerosol dynamics. The deposition (not studied in detail in the current study)
is directly linked to the air concentrations via wet and dry deposition, and development of possible hotspots is directly linked
to the spatial and temporal timing of and the intensity of precipitation events. In this comparison there are no positive percentile
values i.e. including advanced aerosol dynamics always increases wet deposition. With a fixed activation diameter, in
combination with omission of growth processes influencing the carried aerosol number size distribution, there is no way for
the model to replenish the CCN's. Thus, any process providing a mechanism growing the particles below the activation
diameter to a size above it, will enhance the overall deposition.

When simulating dispersion of radioactive material from a nuclear accident with a LPDM this study suggests that the following aspects should be considered:

- It is advisable to know and use best available information regarding current ambient aerosol PNSD onto which the emitted nuclides are condensated. Aspects of the chemical composition might also be important for hygroscopicity and reactivity with other species but that has not been the focus for this study. How the initial size distribution surface properties will impact the deposition field and lifetime can be, to some degree generalized as described in the following paragraphs:

  1.) In general ambient conditions, the surface area is often already dominated by the accumulation mode. Cases where this applies requires an adaptive activation scheme reflecting the competition between cloud droplet nucleation/growth and generation of supersaturation within the cloud. As this is true for a majority of the cases in this study, schemes with fixed activation diameter will always underestimate the wet deposition if all other processes remain constant.

  2.) If the initial carrier aerosol surface size distribution is to a large degree dominated by particles in the coarse mode (1µm and above) the deposition field and lifetime of attached radionuclides attached will largely be controlled by dry deposition via sedimentation and impaction.

  3.) In general ambient conditions, the surface area is often already dominated by the accumulation mode. Cases where this applies requires an adaptive activation scheme reflecting the competition between cloud droplet nucleation/growth and generation of supersaturation within the cloud. As this is true for a majority of the cases in this study, schemes with fixed activation diameter will always underestimate the wet deposition if all other processes remain constant.

- A good air/cloud interaction scheme in LPDMs is of greater importance than the other aerosol dynamic processes when air concentration is the result of the simulations that is of interest. If the sizes of the particles are important, (for dose calculations etc.) description of the evolution of the PNSD is important and then all processes are needed.

- The seasonal variation of the remaining air concentration of released caesium (less in the summer) comes from variations in both initial surface area size distribution and in the amount of precipitation. The interaction between those two effects explains the similarities between the Melpitz and the Zeppelin result even though Melpitz has a higher particle number concentration and Zeppelin has much less precipitation.

- The range of simulated [137]Cs air concentrations for Melpitz and Neo grows with increasing time over the whole 10-day period whereas for Zeppelin the spread grows the first five days and decreases after that, c.f. Fig. 4. In Zeppelin eventually most of the particles are washed out at the end of the 10-day period. As long as there is precipitation that implies wet deposition, all [137]Cs particles will eventually be washed out. The time to wash out depends on the conditions for the individual trajectory, and hence the meteorology, the season, and the geographical location of the release event.

- It is also worth noting that while this study has focused on improving the description of the aerosol dynamical processes (model physics) to be included in an LPDM, the accuracy and the resolution of the numerical weather prediction model is of great importance for dispersion modelling. The numerical weather prediction model governs

the simulation of the trajectories themselves (clouds, precipitation, meteorological parameters and the path involving the distribution of the released material and the interaction with new sources along the way).

To conclude, the PNSD of the ambient aerosol and more advanced aerosol dynamic simulations are important in the case of simulating individual events for example a release from a nuclear power plant. It would be interesting to expand this study with other geographical sites to broaden the knowledge of the impact of aerosol diversity.

The best way to include advanced aerosol dynamics in LPDMs to balance computational cost and the benefits of more detailed results needs to be dealt with in future studies, even though our results put an emphasis on advanced cloud interaction schemes.

The data in this study can also be used for comparisons between different types of trajectories e.g. over sea or over land, close to the surface or high up in the atmosphere etc. This would be an interesting topic for future studies.In this paper we have shown the impact of including more advanced aerosol dynamic parameterizations in dispersion modelling for emergency preparedness. We hope that our results will spur a discussion between decision makers, scientist and modellers in devising the next generation of modelling tools for radiological preparedness

**Data availability**

Data needed to initiate the simulations in this study (three modal log normal fits of the PNSD measurements) are available at PANGAEA, (von Schoenberg et al., 2020a, b, c). Contact corresponding author for more information (pontus.von.schoenbeg@foi.se). Meteorological data needed for the simulations were forward trajectories calculated with HYSPLIT (Stein et al., 2015) and available at https://www.ready.noaa.gov/HYSPLIT.php.

**Author contribution**

PvS, PT, RK and NB designed the study. PvS, PT and HG adjusted the model CALM for this experiment. The simulations were performed and analysed by PvS. AW contributed to the interpretation of the results. The manuscript was drafted by PvS. All authors read and critically reviewed the manuscript.

**Competing interest**

The authors declare that they have no conflict of interest.

**Acknowledgement**

We thank the Swedish Radiation Safety Authority, SSM, and the Swedish ministry of defence for funding this study. We also thanks Swedish Environmental Protection Agency (Naturvårdsverket) for support of observations at Zeppelin station. Observations at NEO were supported by Bolin Centre for Climate Research.

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
