# Peer review of "Aerosol dynamics and dispersion of radioactive particles"

_Atmospheric Chemistry and Physics, 2020_

## Referee Comment (RC1) · Anonymous Referee #1 · 17 Oct 2020

General Comments

In this paper, the authors describe work examining how the consideration of full aerosol dynamics and complex cloud modelling impacts on the prediction of air concentrations of Caesium-137 when compared to the simple scavenging schemes used in Lagrangian particle dispersion models. Their results show that the use of complex cloud modelling results in increased loss of caesium-137 from the atmosphere when compared to simpler cloud schemes. They also show that in most cases other aerosol dynamical processes have a much smaller impact on the air concentration of caesium-137 but in a few cases the impact is much greater.

The aim of this paper is demonstrating the advantages of including advanced aerosol physics in Lagrangian dispersion model. Therefore, I think it would be helpful if the

authors could comment in the discussion on the likely impact of their results if, for example, more complex cloud physics had been used in modelling the Caesium-137 from Fukushima. A number of papers have shown that, at longer range, the air concentration of Caesium-137 from Fukushima is underestimated by Lagrangian models. As the results here show that advanced cloud physics will increase the amount of Caesium-137 lost from the atmosphere it would appear that including advanced cloud physics in a model to simulate Fukushima would result in a greater underestimate of the Caesium-137.

It would aid my understanding of the results if some of the descriptions in the introduction (and method) were made more specific. For example, I would like to see a clearer description of the processes the authors are considering to be advanced aerosol processes and the processes the authors consider as advanced cloud physics. Not all radiological dispersion modellers are familiar with the naming convention for different particle sizes so a list of these would be helpful. Finally, there are a number of locations were information is presented in and order which to me is not intuitive. For example, the paragraph at line 50 introduces aerosol particles yet their physical properties are described in the previous paragraph. Also, the second paragraph of the discussion provides an overview of the different aerosol processes which would have been useful to read in the introduction.

More generally, the method is clearly laid out and the results section is easy to follow. I have a few more specific comments (see below) but I would recommend this paper is published subject to minor revisions.

Specific Comments

The authors used a Lagrangian trajectory model to obtain the path of the air masses through the atmosphere and then a box model to model changes in the properties of the air mass. Since the size of the particles are only considered in the box model and not the trajectory model, I think this means that gravitational settling can only be

considered as a loss process. Many Lagrangian models take gravitational settling into account as a vertical velocity allowing particles to move downwards through different flow fields. Would the authors be able to comment if or how gravitational settling is taken into account and the impact of this assumption?

The end of the paragraph between lines 80 and 86 seems to suggest that Eulerian models are not used in emergency preparedness. I know several centres that do use Eulerian models in emergency preparedness, but I believe that like the Lagrangian emergency models they do not use complex aerosol dynamics. Would the authors be able to rephrase the end of this paragraph to reflect this?

On lines 129-131 the authors describe calculating trajectories using HYSPLIT. Could the authors comment on whether the trajectories were based on mean winds and turbulence or just mean winds?

In Figure 2 for the station at Zeppelin there appears to be a big difference in the initial number size distribution between the winter and summer months. Is this difference expected and if so, what is the cause?

On line 225-226 the authors note that "the effect of ice components in clouds is neglected . . .". Are the authors able to provide an estimate of how much this will overestimate the scavenging efficiency?

Lines 300-302. The authors describe a rapid decrease in the total particle number concentration in the first hours of the simulation. I am not sure I can see this in figure 3 and I am wondering if it occurs so early that it merges into the left-hand x-axis. If this is the case would the authors be able to note this in the text.

Figure 4: The colours in the caption for this figure do not match the colours in the legend although the colours in the legend fit with what is described in the text. I would recommend moving to the colours described in the caption as it is usual to associate warmer colours with the summer months. The authors could also consider making

some of the lines dashed as it is difficult to uniquely pick out 12 different colours.

Line 351: I think this paragraph would benefit from an introductory sentence or two explaining why precipitation is being considered.

Minor Corrections

The following are suggestion for small modifications to improve the readability of the manuscript.

Line 28: Replace "have currently" with "currently have"

Line 33: I think this should be "radiological species" not "radiological spices"

Line 57: Replace "determine also composition" with "determine the composition"

Line 66: Replace "dispersion models for these purposes" with "dispersion models for emergencies"

Line 76: Replace "simulations as both dry and wet" with "simulations of both dry and wet".

Line 81: Remove "that" after reference

Line 158: Add a space after "7178"

Line 179: Replace "external mixture" with "externally mixed" to match "internally mixed" on the previous line

Line 283: Remove "and" after "The outcome"

Line 304/5: Replace "decreases the total particle number concentration" with "the total particle number concentration decreases"

Line 313: Replace "Table 1" with "Table 2"

Line 314: Replace "each of the three stations" with "one of the three stations"

[Figure]

Line 318: Add a bracket after "ALL PROC"

Figure 5: Would benefit from being larger.

Line 359: Remove "the remaining"

Figure 6: As the lines are only constructed from a small number of points, I think it would be helpful to mark those points along the lines.

Page 16: Consider replacing "annual variability" with "monthly variability" or "intra-annual variability". To me, "annual variability" suggests the variation between years not months.

Line 395: Consider starting a new paragraph with the sentence that begins "Figure 7"

Line 401/2: I don't think the lack of December data needs repeating again here.

Line 403: I think this should be "individual order" rather than "individual relation". "Relation" would imply that the gaps between the lines don't change and they do.

Lines 417-420: I'm struggling to read this sentence and would recommend splitting it into two sentences.

Line 445: Replace "and" at the beginning of the line with "during".

Line 446: For emphasis I would suggests starting a new sentence when describing the impact on the deposition field.

Figure 8: I would find it helpful if you could include a sentence in the figure caption highlighting the different x-axis scales.

Line 467: Replace "appearance" with "characteristics

Line 476: Replace "It determines" with "They determine"

Line 485: Replace "slightly" with "slight"

Line 516/517: Replace "where the median" with "the median" and remove "for the" at

the beginning of line 517.

Line 546: Think this should be ">100nm" rather than "<100nm"

Line 559: There is a "t" missing in "precipitation".

Paragraph beginning line 565: There are some brackets missing in this paragraph

Line 603: Remove "there" after "LPDM"

Line 618: Remove repeated "the" and replace "wary" with "vary".

---

## Referee Comment (RC2) · Anonymous Referee #3 · 21 Dec 2020

Recommendation: Major revision

General comments:

The current manuscript developed a coupled detailed aerosol dynamics and cloud processes - Lagrangian trajectory model system and compared the simulation results under the hypothetical nuclear accident situations. The scope of the manuscript is well suited to Atmospheric Chemistry and Physics. Even though the results presented are not surprising (small impacts of ambient aerosol dynamics and large impacts of wet deposition scheme selections), it is worth publication as the manuscript contains some new materials (new model development). However, the current manuscript is not acceptable in its current form because their main arguments are somewhat misleading for the following aspects.

[Figure]

1. A detailed model may not be necessarily a better model or a model closer to truth. Detailed models need many parameters which contain uncertainties. Unless compared with observations, one cannot say which is closer to truth. Sometimes, simple models performed better. The author stated that the detailed cloud model is closer to truth but their method is still far from truth for the following reasons: (1) the in-cloud model only considers warm rain processes but many of the clouds involves cold rain processes in reality in mid-latitudes. (2) The authors used Laakso et al. (2003), but it is not a theoretical but an empirical model. The authors claimed that the first order models used in LPDM are empirical, but the authors also used an empirical model in their below cloud scavenging. How can the authors prove that their method is better? (3) The most critical point is that the cloud process involves vertical motions of air due to convection, but convection cannot be treated by the trajectory method. At least, LPDM needs to be used to say something about the cloud processes. Consequently, at this moment, to the reviewer, the current manuscript is not to show the comparison between the simple and accurate models, but just to show the comparison between two different schemes, simple and complex. (4) Additionally, there are plenty of first-order schemes with different parameters, but the authors selected only one. There could be schemes which predict more depositions than their complex model, or could be ones predict similar values with their complex one.

2. The authors claimed that aerosol dynamics is important for some cases, but compared to what? How is it important with respect to the grid resolution issues, uncertainty in real time forecast due to chaotic nature of atmosphere, and huge variations due to different selections of cloud microphysics modules and other physical modules? The reviewer fully agrees that model development is always important, but the importance varied substantially depending on the processes and properties. It is not fair to state that aerosol dynamics is important, without comparing other issues except wet deposition. (and with compared to the deposition, the impact of aerosol dynamics is negligibly small.) The reviewer also suspects that the presence of coarse mode particles, which was not considered in the model, could be as important as (or even more important

than) aerosol dynamics as presented in the manuscript.

Overall, the reviewer suggests that the context of the manuscript should be as follows: (1) detailed aerosol and cloud schemes are implemented in a trajectory model, (2) aerosol dynamics was less important compared with other processes such as wet deposition and probably meteorological simulations, (3) selection of a good wet deposition scheme is important as the different schemes caused very different results.

Because the authors did not compare their simulation results against any observations in terms of air concentration and deposition of radioactivity and because their wet deposition calculation is still far from reality, the reviewer highly recommends to remove the phrases such as "accurate" or "close to truth". The authors may use a term such as "physically consistent" but the reviewer does not support this as well due to the reasons as listed above: (1) – (3) in general comment #1.

3. To the reviewers, the assumption of 10 days being without exchanges with surrounding air may not be realistic. The authors mainly discussed the simulated difference at 10 days but the differences at 1 day is much more important than that at 10 days. Especially for the emergency situations, pollution near the source is much more important. Probably impacts of aerosol dynamics is much less important for the shorter time integrations, i.e., for the highly contaminated areas near the source. This aspects should also be addressed.

Specific comments:

(1) P.1, ln. 22-23, The last sentence of Abstract, "e.g. in 5% of . . . to a simplified version of the model" is hard to understand. Better to be replaced by clearer statement.

(2) P.2, ln. 47: & -> and

(3) P.2, ln. 49: "deposition, and the potential to act as cloud condensation nuclei": CCN activity may be a part of "deposition", but why is CCN separated from "deposition"?

(4) P.3, ln. 78-79: The sentence "Therefore, the total . . . no dynamic feedbacks on

the aerosol-cloud interaction" needs additional explanation. Aerosol-cloud interaction has various feedbacks such as cloud albedo, cloud life time, and thermodynamic and microphysical invigorations. Which feedback processes can cause underestimation of total wet deposition and how?

(5) P.3, ln. 85: "these Eulerian approaches including advanced aerosol dynamics are however not easily adapted to a Lagrangian framework." It seems not very difficult. There exists Lagrangian – Eulerian hybrid methods. Recently, Danielache et al. (2019) (https://doi.org/10.2343/geochemj.2.0542) coupled Eulerian chemical processes to a LPDM framework, although it does not seem to couple aerosol dynamics processes to the radio-Cs modeling, as done by the current manuscript.

(6) P.4, 2. Method: Horizontal and vertical grid resolutions are missing. How they considered vertical motion of trajectory in HYSPLIT is missing. Brief descriptions of CALM may also be needed such as emission inventories (anthropogenic, biomass burning, biogenic, and volcanic emissions), chemical reactions (gas, aerosol and aqueous phase reactions), and aerosol representations (modal or sectional) used. It may help to grasp how their aerosol dynamics parameters are properly set.

(7) P.5, ln. 134: "gap" -> "missing data" may be better.

(8) P.6, ln. 161-162: "super-micrometre size range is however not included in this study". Aerosol dynamics is less important for coarse mode particles or chemical evolution may enhance the deposition efficiency of coarse mode particles. Neglecting the presence of coarse mode particles in this study may overestimate the impacts of aerosol dynamics processes, which should be noted. If the surface area of coarse mode particles is larger than that of ultrafine particles, gaseous Cs are more condensed to the surfaces of coarse mode particles.

(9) P.6, ln. 170: "Matlab function fmincon.m" Use mathematical expression, rather than a function name of a software.

(10) P. 7, Fig. 2 Units of dN/dlogDp and dS/dlogDp may be missing.

(11) P. 7, ln. 180-181: The sentence "the internal mix of the particles into three different chemical groups, ammonium bisulfate, condensable and partially water-soluble organic vapour and non-condensable insoluble compounds" is awkward. First, "vapour" is not "particles". What is "non-condensable"? Either highly volatile gas or non-volatile particles can be both "non-condensable". Where is ammonium nitrate and ammonium sulfate? Same questions for Table 1 in P. 10.

(12) P. 8, ln. 220-221: What is "column precipitation intensity"? How about the unit?

(13) P.8, ln. 225-226: "The effect of ice components in clouds is neglected, which may overestimate the scavenging efficiency". Why not "underestimate"? Do you assume that the presence of solid hydrometeors delays the scavenging of aerosols? Why?

(14) P. 11, Result -> 3. Results

(15) P. 15, Fig. 6: It is not clear to the reviewer why the difference between purple and green are very large, whereas the difference between yellow and blue are negligibly small. It looks as if "aerosol dynamics are very important when the simple deposition schemes are applied". To my understanding, the simple deposition schemes are independent on aerosol sizes, and thus the difference between purple and green should be even smaller than the difference between yellow and blue.

(16) P. 16, ln. 388: "cloud formation is buffered by newly formed particles" does not make sense. Which buffer system (Stevens and Feingold, Nature, 2009) do the authors indicate? The newly formed particles decrease the maximum supersaturation of air parcel to form cloud?

(17) P. 16, ln. 401: "Melpitz and ." and what?

(18) P. 19, Discussion -> 4. Discussion

---

## Author Comment (AC1) · 1 Feb 2021

Dear editor and reviewers.

We greatly appreciate the constructive and valuable input we have received from the reviewers. Implementing the changes to accommodate your remarks has substantially improved the article. Especially it has made it more focused, the method of the study is clearer which in turn made the conclusions more precise.

We address all comments below and describe the changes that have been made to the article.

**Response to Referee #1, RC1**

**1)** *Comment:*
*The aim of this paper is demonstrating the advantages of including advanced aerosol physics in Lagrangian dispersion model. Therefore, I think it would be helpful if the authors could comment in the discussion on the likely impact of their results if, for example, more complex cloud physics had been used in modelling the Caesium-137 from Fukushima. A number of papers have shown that, at longer range, the air concentration of Caesium-137 from Fukushima is underestimated by Lagrangian models. As the results here show that advanced cloud physics will increase the amount of Caesium-137 lost from the atmosphere it would appear that including advanced cloud physics in a model to simulate Fukushima would result in a greater underestimate of the Caesium-137.*

**Answer:**
We believe that the result of this study show indications that including more detailed aerosol dynamic simulations in LPDMs is motivated based on the results presented in this study.

The question about the best standard for doing this is however not clear. The challenge is how to make an appropriate balance between detailed model physics and available model input in form of relatively coarse weather data (compared to detailed microphysics) and crude assumptions about the source and other parameters. The details in the simulation should not drown in the crudeness of necessary assumptions, generalisations and averaging in time and space.

We have however made an attempt to analyse the effect of including advanced aerosol dynamic simulations compared to the situation where it is not included. It is still a theoretical approach using one model without comparison with measurements. The idea is to compare the model setup in a large number of simulations. We have analysed 23000 different situations (in terms of weather and ambient aerosol description) to span a range of different situations and analyse the statistics. Even though the study considers a Fukushima type of accident it is difficult to compare our experiments with the real Fukushima case and its measurements.

The Fukushima accident is a single realisation with its many difficulties. One was the handling of model data from the NWPs (numerical weather prediction models) to properly describe the precipitation events that was so crucial to the deposition patterns and air concentration patterns. The difficulty to describe the source term in time and space is another. The assumption of what kind of radioactive aerosol that was created is also important. Ideally one would use detailed information about the ambient background aerosol, highly resolved aerosol dynamics, detailed cloud activation and wet removal schemes as well as high resolution meteorological information. As an example, (Sato et al., 2018) simulated dispersion from the Fukushima accident on a highly resolved weather using standard dispersion models, whereas we use a highly resolved aerosol dynamic model on standard weather.

It is difficult to compare median result of 23000 simulations with one real life case, the Fukushima case. The result of the statistics show that for the median, the inclusion of advanced aerosol dynamics decrease the air concentration and increase the wet deposition in most cases, when it comes to the simplified wet deposition scheme (FAS, Fixed Activation Size), see Fig. 8 right column. In the case of

the activation size calculated from updrafts, Fig. 8 left column, there are more cases when advanced aerosol dynamics increase air concentration instead. The median is closer to zero and there are percentiles that are both positive and negative. The fixed activation size, FAS is also a different approach than using a general wash out parameter, commonly used in LPDMs, since also this method includes the partaking in cloud formation of the particles making the wet deposition dependant of particle size.

**2)** *Comment:*

*It would aid my understanding of the results if some of the descriptions in the introduction (and method) were made more specific. For example, I would like to see a clearer description of the processes the authors are considering to be advanced aerosol processes and the processes the authors consider as advanced cloud physics. Not all radiological dispersion modellers are familiar with the naming convention for different particle sizes so a list of these would be helpful. Finally, there are a number of locations were information is presented in and order which to me is not intuitive. For example, the paragraph at line 50 introduces aerosol particles yet their physical properties are described in the previous paragraph. Also, the second paragraph of the discussion provides an overview of the different aerosol processes which would have been useful to read in the introduction.*

**Answer:**

Following the referee #1 recommendation we have added a more detailed description about the aerosol processes considered, but refer the reader to other sources for an in-depth description e.g. (Seinfeld and Pandis, 2006). Specifically we have added information of aerosol processes in the introduction. There we have also included a figure with the naming conventions of different particle sizes.

P.2, ln. 44-56 has been rewritten from:

[revised manuscript text omitted]

After the sentence *"… is the most important sink for atmospheric aerosols."* P.2, ln. 60 the following sentence has been added:

There are many good sources for in depth information concerning aerosol physiochemical processes and atmospheric aerosols e.g. (Seinfeld and Pandis, 2006).

A parenthesis has been added in the sentence on P.4 ln. 111-113 to emphasis what we in this context consider advanced aerosol dynamics.

The sentence has been altered from:

*"The main purpose of this study is to evaluate how detailed treatment of aerosol dynamics change atmospheric lifetime and deposition of radioactive material compared to the standard way currently adopted in LPDMs i.e. treating deposition as a first order process only."*

To:

"The main purpose of this study is to evaluate how detailed treatment of aerosol dynamics (including condensational growth, coagulation, nucleation, partaking in cloud formation as CCN, chemical transformation, and interaction with new sources) propagates to atmospheric lifetime and deposition efficiency of radioactive material compared to the standard way currently adopted in LPDMs i.e. treating deposition as a first order process only."

A description of parameterization in the full scale model of the in-cloud physics is done in chapter 2.4 Model setup. To make it clear that this is what is referred to as "advanced wet deposition" or "advanced cloud parameterization" later in the MS the description of experiment no 3 has on P.10, ln.166-169 has been changed from:

*"In this experiment however, the wet deposition scheme is more advanced than in many LPDMs, including below-cloud scavenging and in-cloud scavenging with CCN activation based on updrafts and available humidity. This experiment is mimicking the behaviour of a LPDM with an advanced wet deposition scheme. Abbreviation: ONLY DEP"*

To:

"In this experiment however, the wet deposition scheme is more advanced than in many LPDMs using a scavenging coefficient. It includes below-cloud scavenging and in-cloud scavenging with CCN activation based on updrafts and available humidity (henceforth denoted advanced wet deposition scheme or advanced cloud parameterization)."

In the next segment about experiment 4 a similar description has been made about what later in the MS will be referred to as "advanced aerosol dynamics". P. 10, ln. from:

*"This experiment represents the behaviour of a dispersion model with classic wet deposition scheme, but where the advanced aerosol dynamics is included."*

To:

"This experiment represents the behaviour of a dispersion model with a simplified wet deposition scheme, but where the coagulation, condensation, emissions, chemical transformations and nucleation (henceforth denoted advanced aerosol dynamics) is included."

Referee #1 Specific Comments:
3) *Comment:*
*The authors used a Lagrangian trajectory model to obtain the path of the air masses through the atmosphere and then a box model to model changes in the properties of the air mass. Since the size of the particles are only considered in the box model and not the trajectory model, I think this means that gravitational settling can only be considered as a loss process. Many Lagrangian models take gravitational settling into account as a vertical velocity allowing particles to move downwards through different flow fields. Would the authors be able to comment if or how gravitational settling is taken into account and the impact of this assumption?*

**Answer:**
It is correct that gravitational settling in this case is only a loss process that does not change the path of the model particle. Furthermore, in our setup the loss process is only active when the trajectory is within the mixing layer. This neglects the loss of particles due to gravitational settling when the trajectory above the mixing layer. The method comes from how the Lagrangian Trajectory model is

adapted to this study and we consider this to be of lesser importance for the conclusions of this study. If ideas from this study should be implemented in Lagrangian Particle Dispersion Models it has to be addressed properly. A new paragraph has been added after P.8, ln.208:

"The dry deposition includes Brownian diffusion, most effective for particles smaller than 50 nm, and gravitational settling and inertial impaction, which is most effective for particles larger than a few micrometres in size. Dry deposition in the model setup is only active when the box is in the mixing layer, when particles can reach the ground. This neglects the effect of gravitational settling for particles above the mixing layer and this effect can be large when aerosol surface is dominated by coarse mode particles.

**4) Comment:**
*The end of the paragraph between lines 80 and 86 seems to suggest that Eulerian models are not used in emergency preparedness. I know several centres that do use Eulerian models in emergency preparedness, but I believe that like the Lagrangian emergency models they do not use complex aerosol dynamics. Would the authors be able to rephrase the end of this paragraph to reflect this?*

**Answer:**
We agree that it was not correctly phrased and has adjusted the end of the paragraph:

The original version:

*"For emergency preparedness where fast models are essential these Eulerian approaches including advanced aerosol dynamics are however not easily adapted to a Lagrangian framework."*

Has been changed to:

"For emergency preparedness fast models are essential. To our knowledge there has been no solution presented providing sufficiently fast and detailed aerosol dynamics representations within the strictly Lagrangian framework applied in LPDM's used for dispersion simulations of radionuclides"

An additional sentence about an Eulerian-Lagrangian hybrid model has been added in response to referee #3, specific comment no 5. further down in this document.

**5) Comment:**
*On lines 129-131 the authors describe calculating trajectories using HYSPLIT. Could the authors comment on whether the trajectories were based on mean winds and turbulence or just mean winds?*

**Answer:**
We fully agree and have clarified those lines by replacing the original:

*"The trajectories used in this study were calculated with the model HYSPLIT in forward mode (Stein et al., 2015) using fields 130 from the meteorological model GDAS (NOAA 2019). 2019). From these calculations, both the trajectory path and important meteorological parameters along the path was retrieved and used as input for CALM."*

with:

"The three dimensional (latitude, longitude and height) trajectories used in this study were calculated with the model HYSPLIT in forward mode (Stein et al., 2015) using fields from the meteorological model GDAS (NOAA 2019) with a geographical resolution of 1 deg in latitude and longitude and

temporal resolution of 3 hours. From these calculations, both the trajectory path (based on 3-dimensional mean winds) and important meteorological parameters along the path was retrieved and used as input for CALM."

**6)** *Comment:*

*In Figure 2 for the station at Zeppelin there appears to be a big difference in the initial number size distribution between the winter and summer months. Is this difference expected and if so, what is the cause?*

**Answer**:

To clarify this the following section has been added after P.6 ln. 172:

"Seasonal variations in the PNSD can be noticed for all stations, where Zeppelin stands out, c.f. Fig 2. PNSD seasonality in the artic is driven by both local sources, remote sources and transport patterns as well as differences in precipitation patterns, (Tunved et al., 2013). The total concentration of particles is lower in the arctic than for the other station which makes it more sensitive for variations. The spring period, February to April, consists of an aged and elevated accumulation mode aerosol strongly influenced by remote sources linked to meteorological transport patterns and inversions trapping the aerosol and reducing dilution. This is commonly referred to as *Arctic Haze.* The number concentration during the summer period, May-September, is formed through mainly gas-to-particle conversion resulting in a pronounced Aitken mode towards as well as intermittent presence of a nucleation mode, c.f. Fig 2. Since particle formation is dependent of sunlight, there are also strong diurnal variations in this period."

*Comment:*

*On line 225-226 the authors note that "the effect of ice components in clouds is neglected . . .". Are the authors able to provide an estimate of how much this will overestimate the scavenging efficiency?*

**Answer:**

We agree with the referee. Since we do not know the magnitude of the impact of ice components and we do not know if the effect will result in over or under estimations there of we adjusted the sentence on p 8 ln. 225-226.See also answer outlined under specific comment 13, Referee #3:

Original version:
*"The effect of ice components in clouds is neglected, which may overestimate the scavenging efficiency."*

Changed version:

"The effect of ice components in clouds is neglected, which may alter the scavenging efficiency and ultimately the wet removal rate."

**7)** *Comment:*

*Lines 300-302. The authors describe a rapid decrease in the total particle number concentration in the first hours of the simulation. I am not sure I can see this in figure and I am wondering if it occurs so early that it merges into the left-hand x-axis. If this is the case would the authors be able to note this in the text.*

**Answer:**

The Referee is right, this was wrongly phrased and is adjusted to the following that is in accordance with what can be seen for Neo in figure 3. From:

*"The total particle number concentration decrease very quickly within the first hours of simulation."*

To:
"The total particle number concentration decreases after a small peak half a day into the simulation"

**8) *Comment:***
*Figure 4: The colours in the caption for this figure do not match the colours in the legend although the colours in the legend fit with what is described in the text. I would recommend moving to the colours described in the caption as it is usual to associate warmer colours with the summer months. The authors could also consider making some of the lines dashed as it is difficult to uniquely pick out 12 different colours.*

**Answer:**

To make this picture clearer we have followed the referees appreciated comment by adding dashed lines as well. We agree that the colour references in the caption could improve and that has been adjusted by correcting and clarifying the caption accordingly. The new figure looks like this:

[Figure]

The figure caption has been changed from:

"Figure 4: Annual (black) and monthly means (shades of blue and green, starting with blue for the winter months, Dec, Jan and Feb). Development of caesium concentration as normalized Becquerel for the five different experiments. Each row represent the different stations (from the top, Melpitz, Neo and Zeppelin) and the columns the different experiments, 350 ALL PROC, ONLY DRY, ONLY DEP, ALL PROC FAS and ONLY DEP FAS."

To:

"Figure 1: Annual (black) and monthly means (blue for winter, Dec, Jan and Feb, green for spring, Mar, Apr and May, red for summer, Jun, Jul and Aug and orange for autumn, Sep, Oct, Nov). Development

of caesium concentration as normalized Becquerel for the five different experiments. Each row represents the different stations (from the top, Melpitz, Neo and Zeppelin) and the columns the different experiments, ALL PROC, ONLY DRY, ONLY DEP, ALL PROC FAS and ONLY DEP FAS."

**9)** *Comment:*
*Line 351: I think this paragraph would benefit from an introductory sentence or two explaining why precipitation is being considered.*

**Answer:**
We agree with the Referee and have added a sentence to put the precipitation into context before P.14 ln. 351:

"Wet deposition is the by far the most efficient removal process for accumulation mode particles due to slow diffusion and terminal velocity. For reference, the total accumulated precipitation for each trajectory is shown in Fig. 5."

**Referee #1 Minor corrections**
Below we outline our response to the minor corrections raised by Referee #1. The Referee comments are shown in italics, response in regular font and changes in the MS in red.

*Line 28: Replace "have currently" with "currently have"*
We have corrected accordingly.

*Line 33: I think this should be "radiological species" not "radiological spices"*
We have corrected accordingly.

*Line 57: Replace "determine also composition" with "determine the composition"*
We have corrected accordingly.

*Line 66: Replace "dispersion models for these purposes" with "dispersion models for emergencies"*
We agree that this should be rephrased and have replaced the original sentence:

*"For practical reasons, in particular computational limitations and the complexity of physical parameterizations, dispersion models for these purposes have been developed based on simplified physics."*

with:
"Due to computational limitations and the complexity of physical parameterizations, dispersion models for emergency preparedness have been developed based on simplified physics."

*Line 76: Replace "simulations as both dry and wet" with "simulations of both dry and wet".*
We have corrected accordingly.

*Line 81: Remove "that" after reference*
We have corrected accordingly.

*Line 158: Add a space after "7178"*
We have corrected accordingly.

*Line 179: Replace "external mixture" with "externally mixed" to match "internally mixed" on the previous line*
 We have corrected accordingly.

*Line 283: Remove "and" after "The outcome"*
 We have corrected accordingly.

*Line 304/5: Replace "decreases the total particle number concentration" with "the total particle number concentration decreases"*
 We have corrected accordingly.

*Line 313: Replace "Table 1" with "Table 2"*
Thank you that was a mistake, *we have corrected accordingly*

*Line 314: Replace "each of the three stations" with "one of the three stations"*
 We have corrected accordingly.

*Line 318: Add a bracket after "ALL PROC"*
 We have corrected accordingly.

*Figure 5: Would benefit from being larger.*
Figure 5 has been made larger. Also the title and y-labels have been changed to better explain the content:

[Figure]

*Line 359: Remove "the remaining"*
 We have corrected accordingly.

*Figure 6: As the lines are only constructed from a small number of points, I think it would be helpful to mark those points along the lines.*
We agree and have added bullets to the lines in the figure:

[Figure]

*Page 16: Consider replacing "annual variability" with "monthly variability" or "intra- annual variability". To me, "annual variability" suggests the variation between years not months.*
We agree that annual is not a good definition for the variation within a year. We have replaced *"annual variations"* with "seasonal variations" and *"annual variability"* with "seasonal variability". It has been made on P.6. ln. 155, P. 16, ln. 401, ln. 395-396, 398 and 401, In the figure caption for Figure 7 on P.17 ln 408, P. 23, ln. 565, 567, 573 and 580, P. 25, ln 627. On P.25, ln. 631 the term *"annual spread"* has been changed to "seasonal spread".

*Line 395: Consider starting a new paragraph with the sentence that begins "Figure 7"*
We agree and have corrected accordingly

*Line 401/2: I don't think the lack of December data needs repeating again here.*
We agree and have removed that sentence.

*Line 403: I think this should be "individual order" rather than "individual relation". "Relation" would imply that the gaps between the lines don't change and they do.*
We agree and have corrected accordingly

*Lines 417-420: I'm struggling to read this sentence and would recommend splitting it into two sentences.*

We agree that it is a difficult sentence and it has been split into two sentences:

The original:

*"This trajectory-difference represents the impact that advanced aerosol dynamics has when added to a model using dry deposition and advanced cloud parameterization with wet deposition: coagulation, condensational growth, nucleation and interaction with the background aerosol involving new sources then creates the deviation from the 0-line in these plots."*

The new version:

"This trajectory-difference represents the impact that advanced aerosol dynamics has, when added to a model using dry deposition and advanced cloud parameterization with wet deposition. The added processes include coagulation, condensational growth, nucleation and interaction with the background aerosol involving new sources then creates the deviation from the 0-line in these plots."

*Line 445: Replace "and" at the beginning of the line with "during".*
We agree and have corrected accordingly

*Line 446: For emphasis I would suggests starting a new sentence when describing the impact on the deposition field.*
We agree and has made two sentences instead:

Original version:

*"The simulations mimicking a LPDM over predict the air concentration of released caesium that in turn will under predict the deposition field."*

New version:

"The simulations mimicking a LPDM over predict the air concentration of released caesium. This will in turn under predict the deposition field."

*Figure 8: I would find it helpful if you could include a sentence in the figure caption highlighting the different x-axis scales.*

We are not sure what the reviewer referred to in this comment since the x-axis scales are all the same. To make interpretation of Figure 8 easier two sentences has been added in the caption, we hope this addresses the referee's request:

"Negative values represent when ALL PROC has lower concentration than the compared experiment (ONLY DEP, left and ONLY DEP FAS, right) i.e. when advanced aerosol dynamics gives lower air concentrations. Note that the scales of the y-axes vary between the sub plots."

*Line 467: Replace "appearance" with "characteristics*
We have corrected accordingly.

*Line 476: Replace "It determines" with "They determine"*
We have corrected accordingly.

*Line 485: Replace "slightly" with "slight"*
We have corrected accordingly.

*Line 516/517: Replace "where the median" with "the median" and remove "for the" at C5 the beginning of line 517.*
We have corrected accordingly

*Line 546: Think this should be ">100nm" rather than "<100nm" Line 559: There is a "t" missing in "precipitation".*
That is correct and we have corrected accordingly

*Paragraph beginning line 565: There are some brackets missing in this paragraph*
We have corrected accordingly

*Line 603: Remove "there" after "LPDM"*
We have corrected accordingly

*Line 618: Remove repeated "the" and replace "wary" with "vary".*
We have corrected accordingly

**Response to Referee #3, RC2**

**1)** *Comment:*
*A detailed model may not be necessarily a better model or a model closer to truth. Detailed models need many parameters which contain uncertainties. Unless compared with observations, one cannot say which is closer to truth. Sometimes, simple models performed better. The author stated that the detailed cloud model is closer to truth but their method is still far from truth for the following reasons: (1) the in-cloud model only considers warm rain processes but many of the clouds involves cold rain processes in reality in mid-latitudes. (2) The authors used Laakso et al. (2003), but it is not a theoretical but an empirical model. The authors claimed that the first order models used in LPDM are empirical, but the authors also used an empirical model in their below cloud scavenging. How can the authors prove that their method is better? (3) The most critical point is that the cloud process involves vertical motions of air due to convection, but convection cannot be treated by the trajectory method. At least, LPDM needs to be used to say something about the cloud processes. Consequently, at this moment, to the reviewer, the current manuscript is not to show the comparison between the simple and accurate models, but just to show the comparison between two different schemes, simple and complex. (4) Additionally, there are plenty of first-order schemes with different parameters, but the authors selected only one. There could be schemes which predict more depositions than their complex model, or could be ones predict similar values with their complex one.*

**Answer:**
We agree with the referee completely. Added complexity is not a guarantee for a more accurate representation of transport and fate of particles. As the reviewer also says, it is of importance to put any additional level of detail implemented in the light of all the various aspects of process and transport descriptions. Performance can never be better than allowed by the individual process representations, just as the weakest link-analogy. It is however not futile to study how changes of process descriptions propagates to changes in the model result. This is how we learn where we can cut corners, and where we cannot.

The main goal of this study is to investigate how implementation of a detailed representation of ambient aerosol dynamics and cloud activation scheme influence the transport potential and lifetime of radionuclides as compared to model designs where approaches that are more rudimentary are applied. It is all well and true that we really cannot say anything about how either of these pathways, detailed versus simplified, will lead to a better model result. The only thing we can say is whether detailed descriptions substantially will change the results compared to the simplified case. If we can show that ambient aerosol dynamics in fact play a large role for simulated results, it is probably a good idea to have a closer look on potential approaches considering these processes.

It is well established, that the lifetime of submicron aerosol mass is controlled by wet removal in general and in-cloud scavenging and removal in particular. One important goal of this study is to investigate how condensation growth, and to a lesser degree coagulation moves particle associated trace compounds into the accumulation mode thus making them available to in cloud removal. The hypothesis was that if this process is rapid, we could potentially disregard aerosol dynamics at the initial stage, leaving us with wet removal as the governing process. If the opposite would be true, we would need to take into account also ambient aerosol dynamics. In the idealized case, the accumulation mode is replenished by condensation growth. Furthermore, the number of particles in the accumulation mode will determine what will be the lower size of activated particles when a cloud forms under any given set of conditions. This is perhaps one of the more central questions we want to pursue in this study.

In light of the above, the study should therefore not be regarded as an attempt to provide a more accurate type of model, but instead build understanding on how ambient dynamics can control rate of wet removal (which, again, is the ultimate sink for submicron mass) in different environments. This is as we have seen not trivial. From this point of view, it becomes clear that we do not try to reach closure, or postulate a best approach. Rather, we investigate under what circumstances, if any, ambient dynamics to some degree could control the lifetime of a tracer.

Specifically, to address the four main points we under "Comment 1" we outline following answers:

1.) We are aware of the fact that considering liquid phase clouds only is a limiting factor, and certainly the modelling exercise would have been more relevant if mixed phase clouds would have been considered. However, the results would have an additional layer of complexity that not necessarily would be beneficial for the outcome of the study.
2.) Laakso et al. (2003) is an empirically based process description, but with main focus on below cloud scavenging. We are mainly targeting in-cloud removal. It would be a nice study to use the same framework as the current one to study the relative roles of below and in-cloud scavenging, but this has not been the focus of the study.
3.) We agree. A trajectory model cannot be used to accurately describe the full complexity of cloud formation. We can however use it in combination with a cloud droplet activation scheme to at least partly assess aerosol cloud interactions. The method, with its limitations, is however deemed sufficient to address the overarching question regarding the role of ambient aerosol dynamical processes.
4.) We agree here as well. However, considering all potential process descriptions is clearly beyond the scope of the study. We acknowledge the potential limitations in the present, but this fact does not render the current study irrelevant.

To make it more clear that the study is not investigating a new dispersion model but analysing the effect of simulating more detailed aerosol processes, changes has been made throughout the MS:

In the abstract the sentences on P.1 ln. 16-18 we have changed the original sentences:

*"The objective for this study is to analyse the importance of including more advanced aerosol dynamic processes in LPDM simulations for the use in radioactive preparedness. In this investigation, a fictitious NPP failure, commencing with hourly separation for a full year, is studied for three geographically and atmospherically different sites."*

To:

"The objective for this study is to analyse the importance of these advanced aerosol dynamic processes if they were to be included in LPDM simulations for the use in radioactive preparedness. In this investigation, a fictitious nuclear power plant failure is studied for three geographically and atmospherically different sites. The incident was simulated with a Lagrangian single trajectory box model with a new simulation for each hour throughout a year to capture seasonal variability of meteorology and variation in the ambient aerosol."

The second sentence in the introduction has been removed since it could be misleading P.1 ln. 25-27. Removed sentence:

*Our target of interest is to investigate a novel and more accurate way of estimating dispersion of radioactive particles following a nuclear power plant failure.*

To further clarify the study and its purpose P.1 ln. 29-P.2 ln. 32 has been modified from:

*"The purpose of the study is to investigate whether aerosol micro-physical processes are important for transport and deposition patterns following a release of radionuclides, study what effects microphysics will have on aerosol lifetime and indirectly on deposition fields in the framework of currently adopted dispersion modelling techniques."*

To:

"The purpose of this study is to investigate whether detailed aerosol micro-physical processes are important when simulating the transport and deposition patterns following a release of radionuclides from a nuclear power plant failure. The aim of the study is to investigate the potential effects that could result from inclusion of detailed aerosol microphysics in dispersion models. Can these processes change simulated aerosol lifetime and deposition fields and are these effects of a high enough degree to encourage implementation these process descriptions into the framework of currently adopted dispersion modelling techniques?"

To make it more clear how the study was executed we have rewritten p.4 ln. 105-107 and added a sentence.

In original MS p.4 ln. 105-107:

"In this way, the used model setup does not simulate a single particle, or a static number size distribution, as the model particles in a classical LPDM, but allow each trajectory to carry a complete PNSD that will age and transform during transport"

Is replaced/modified as:

"In this way, the used model setup does not simulate a single particle, or a static number size distribution, i.e. the most commonly used designs in classical LPDM models. Instead we allow the simulated trajectories to carry a complete PNSD that will age and transform during transport. Simulations have been done for different sites and under different meteorological conditions to account for the variations that can occur due to different weather conditions, different air masses and new aerosol sources along the path. They have all been initialised with a measured PNSD unique for that time and location."

In the Method section P.4 ln. 123-125 the sentences:

*"A full year of PNSD measurements of the ambient aerosol at each station initiated the model. Different experiments were performed for all the trajectories to investigate the importance of simulating different processes."*

has been adjusted to:

"A full year of PNSD measurements of the ambient aerosol at each the three stations initiated the model for each individual simulation. Different modelling experiments were performed for all the trajectories to investigate the importance impact of simulating different processes."
to make the method and purpose more clear.

To continue making the method clearer the first sentence in 2.1, P. 4 ln. 127-128 the sentence:

*"By using 24 different trajectories each day (one for each hour) the variability of the ambient PNSD, in the meteorology and transport patterns could be analysed"*

has been changed to:

"By simulating 24 different trajectories each day (one for each hour) the variability of the ambient PNSD, in the meteorology and transport paths could be analysed."

On P. 5 ln. 134 "they" has been replaced by "the"

On P.6, ln. 132, the sentence:
*"For each trajectory, CALM was run for 10 days for different  experiments"*

has for clarity been changed to:

"In each individual simulation, CALM is run along the air mass trajectory for the duration of 10 days"

To after P.8, ln.208:

For clarity, we expanded the description of how we treat clouds in the model, and added the following section under section 2.4 after the addition made do address referee #1 3rd comment (the first of the specific comments).

The  following segment is included on P. 8 before the paragraph that starts, "Concerning wet deposition …" on P.8 Ln. 209:

"The model design utilizes a hybrid approach and considers two different compartments: one for ambient aerosol dynamics and one for aerosol cloud interactions and in-cloud scavenging. The dry "ambient dynamic" box considers detailed descriptions of gas-to-particle-conversion, dry deposition, and coagulation. Decoupled from this box we run an adiabatic 1-dimensional cloud module that calculates activation and growth of aerosol particles in an ascending air parcel. The cloud model is run separate from the ambient dynamics box when clouds are prescribed. The cloud compartment results in a droplet distribution of the activated particles, which in turn allow us to calculate a liquid water content (LWC).  The environmental parameters framing the cloud calculations are based on meteorological parameters provided by the meteorological model (GDAS) which are calculated every

3h. Exactly how this is done is outlined below. Once formed, available SO2 is equilibrated to the bulk water together with ozone and hydrogen peroxide. This allows for calculation of pH and concentration dependent liquid  phase oxidation of sulfur dioxide. This means that if the cloud does not precipitate, the sulfate produced through this pathway is distributed over the activated particles. This means that the cloud does in fact have the potential to act as a source of aerosol mass.

Apparently, there is a discontinuity comparing on the one hand the cloud, and on the other hand the ambient box. We have chosen this approach since we want to retain the key processes of activation and its link to aerosol size distribution properties and chemistry. Once the cloud dissipates, the effect it has had on the aerosol in the ambient dynamics box (in-cloud chemistry and in-cloud scavenging) is evaluated based on the fraction of the box that has been influenced by the cloud, which is determined from humidity profiles and fractional cloud cover."

In the section "Methods" (Sect. 2.5 The different experiments P. 10 ln. 253-254) the introduction before the bullet list has been changed and extended from:

*"For each of the trajectories we made five different experiments to analyse the importance of aerosol dynamics, summarized in Table 2."*

To:
For each of the trajectories we made five different experiments simulating single trajectories to analyse the impact of aerosol dynamics, summarized in Table 2. The experiments were designed in a way to, as transparently as possible, evaluate to what degree simulating advanced aerosol dynamics in LPDMs could influence the overall result. Identifying where and when detailed treatment could be beneficial is important in the context of emergency preparedness after a nuclear power plant accident:"

To increase coherence and readability *"aerosol number size distribution"* and *"particle number size distribution"* has been replaced by "PNSD" on P.5, ln. 140 and P.20, ln. 479.

The word "importance" has been changed throughout the article to "impact" where it concerns the impact that aerosol dynamics have the simulations. This is more accurate to the layout of the study and is in line with the comments of the referee. This has been done on P.4. Ln.121, P.6, ln. 141, P.8, ln. 208, P.10, ln.153, P.20, ln. 403.

2) *Comment:*
*The authors claimed that aerosol dynamics is important for some cases, but compared to what? How is it important with respect to the grid resolution issues, uncertainty in real time forecast due to chaotic nature of atmosphere, and huge variations due to different selections of cloud microphysics modules and other physical modules? The reviewer fully agrees that model development is always important, but the importance varied substantially depending on the processes and properties. It is not fair to state that aerosol dynamics is important, without comparing other issues except wet deposition. (and with compared to the deposition, the impact of aerosol dynamics is negligibly small.) The reviewer also suspects that the presence of coarse mode particles, which was not considered in the model, could be as important as (or even more important than) aerosol dynamics as presented in the manuscript.*

*Overall, the reviewer suggests that the context of the manuscript should be as follows: (1) detailed aerosol and cloud schemes are implemented in a trajectory model, (2) aerosol dynamics was less important compared with other processes such as wet deposition and*

*probably meteorological simulations, (3) selection of a good wet deposition scheme is important as the different schemes caused very different results.*

*Because the authors did not compare their simulation results against any observations in terms of air concentration and deposition of radioactivity and because their wet de- position calculation is still far from reality, the reviewer highly recommends to remove the phrases such as "accurate" or "close to truth". The authors may use a term such as "physically consistent" but the reviewer does not support this as well due to the reasons as listed above: (1) – (3) in general comment #1.*

**Answer:**

This is agreeable a valid point if the purpose of the study was to provide a new model, ready-to-use. We are aware that several assumptions made in the current study potentially could cause deviations much larger than those caused by including ambient dynamics on/off, or by utilizing different choices of activation and aerosol removal schemes. This statement can be valid for just about any change in level of detail for any model. We are rather certain, that there exists nothing as an ideal or flawless model. If it did, any model study would be superficial.

The purpose of the study (as outlined under Comment 1 above) is not to provide a best solution. The study targets ambient dynamics and parameterized cloud activation and investigate under which circumstances they likely can (or cannot) be disregarded. The findings will assist LPDM modellers with input on what direction they would choose when upgrading or otherwise modifying the model park. Can dynamics be disregarded or could they in fact be beneficial to the overall result?  We are not aware of any study that explicitly target this issue, and although admittedly incomplete (as any model exercise) the study could provide insights to field and perhaps spur new and differently designed model experiments targeting the same issue. How to implement it in a computational efficient manner is still an open question, though.

Regarding the coarse mode, the referee is right that initial attachment of nuclides to coarse mode particles potentially can give very different results due to rapid sedimentation. This will be, on average, highly dependent on season and geographic location of model initialization as coarse mode particles show high variability in space and time due to the short residence time. In designing the study we did however target the submicron fraction since this is the size range where aerosol dynamical processes play their biggest roles. We acknowledge in the revised MS that including also coarse mode can change the result substantially.

We agree with the Referee that the context should better be highlighted, and changes has been done to the MS accordingly. We agree that the use of words like "accurate" or "closer to truth" is not appropriate. Where applicable, similar phrasings have been replaced using more adequate wording.

Suggestions by the referee has involved many global changes on the MS but the main change is present in the revision of section: Conclusions P. 24 ln. 598 and onwards, outlined in detail under reviewer #3 Comment 3 below.

P.10 ln. 255-257 has therefore been changed from:
*"This simulation represents the representation closest to the truth: what a single trajectory of a LPDM dispersion model run would give as a result if it were to simulate all aerosol dynamic processes."*

To:
"This simulation represents the most detailed description of what a single trajectory of a LPDM dispersion model run would give as a result if it were to simulate all aerosol dynamic processes."

P.21 ln. 500 has for the same reason been changed from:
*"…experiment closest to the truth (ALL PROC), Fig. 7."*

To:
"…the experiment simulating all aerosol dynamic processes. (ALL PROC), Fig. 7"

In P.25, ln. 244-245 the word *"accurate"* has been replaced by "detailed" and the sentence now reads:

*"The best way to include advanced aerosol dynamics in LPDMs to balance computational cost and the benefits of more detailed results needs to be dealt with in future studies, even though our results put emphasis on advanced cloud interaction schemes."*

In general, we have tried to reformulate the revised MS to better highlight what the purpose is and what the simulation results suggest. We do agree with the Referee that some statements in the MS appear misleading as regarding the true goal of the study. We are happy that these can be removed from the MS through the insightful guidance of the referee.

**3) Comment:**
   *To the reviewers, the assumption of 10 days being without exchanges with surrounding air may not be realistic. The authors mainly discussed the simulated difference at 10 days but the differences at 1 day is much more important than that at 10 days. Especially for the emergency situations, pollution near the source is much more important. Probably impacts of aerosol dynamics is much less important for the shorter time integrations, i.e., for the highly contaminated areas near the source. These aspects should also be addressed.*

**Answer:**
We fully agree. For the current study, this does however not play a central role. As better clarified in the manuscript, the purpose of the study is to investigate the potential role of omission of ambient aerosol dynamics and detailed cloud activation schemes: Are simplifications used in current LPDM's (due to the very nature of this kind of models), providing significantly different results compared to a situation where the aerosol dynamics is treated at a higher level of detail? In that context it does not really matter if the trajectories become more inaccurate as simulation time increase. We do however acknowledge that mixing of air surrounding the simulated parcel can serve as a way of replenishing (diminishing) aerosols in the model without the introduction of additional sources (sinks).

In the revised MS on P.20, ln. 467 the following segment has been changed from:

*"The appearance of the initial aerosol PNSD also determines the effect that the different aerosol processes will have. The source fields that the trajectory box model travels through determine, together with the sinks, what new non-radioactive particles the radioactive aerosol will interact with in the simulated box. This will make the PNSD differ from the 137Cs activity size distribution."*

To:
"The characteristics of the initial aerosol PNSD also determines the effect that the different aerosol processes will have. Further, this will also connect to how the radionuclides initially are described. In our study we have assumed that the nuclides are emitted as a low volatile vapour. Other assumptions

could be emission of a pure combustion aerosol that dynamically will interact with the background aerosol. The nature of emission scenario will of course therefor also affect the outcome of the study. The source fields that the trajectory box model travels through determine, together with the sinks, what new non-radioactive particles the radioactive aerosol will interact with in the simulated box. This occurs in this model setup when the box is within the mixing layer (depending on the trajectory path) which will make the PNSD differ from the $^{137}$Cs activity size distribution. We have not studied the complexity near the source within the first day where aerosol dynamic processes might have a different impact, especially when it comes to releases in highly polluted environment. To address the situation near the source other type of dispersions models are more suited for purpose than LPDMs which is not the scope of this study. The focus here has been on the result during and at the end of the 10-day simulation to understand the role of aerosol dynamics on this time scale."

Finally, to properly address all comments from Referee #3 we have rewritten much of the conclusions on P. 24, ln. 598 – P. 25, ln. 649 as follows:

[revised manuscript text omitted]

**Specific comments:**

Below we outline our response to the more specific comments and concerns raised by Referee #3. The Referee comments are shown in italics and response in regular font.

(1) Comment:
*P1. ln. 22-23, The last sentence of Abstract, "e.g. in 5% of . . . to a simplified version of the model" is hard to understand. Better to be replaced by clearer statement.'*

Answer:
We agree and reformulate as follows:

"…we show that inclusion of detailed ambient aerosol dynamics can play a large role for the model result in simulations that adopt a more detailed representation of aerosol cloud interactions. The results highlights a potential necessity for implementation of more detailed representation of general aerosol dynamic processes into LPDM's in order to cover the full range possible environmental characteristics that can apply during a release of radionuclides into the atmosphere."

*(2)  Comment:*
> *P.2, ln. 47: & -> and*

**Answer:**
This has been changed accordingly.

*(3)  Comment:*
> *P.2, ln. 49: "deposition, and the potential to act as cloud condensation nuclei": CCN activity may be a part of "deposition", but why is CCN separated from "deposition"?*

**Answer:**
In the activation scheme applied the model (i.e. estimates of CCN activation under different environmental conditions), activation is not the same as removal. The following deposition is only prescribed if the cloud precipitates. This removal due to in-cloud scavenging is based on calculated lower activation radius, cloud thickness, LWC, cloud fraction and precipitation intensity. This means, that the wet removal not necessarily removes all the activated particles, but instead this amount is scaled following the relations between precipitation intensity and liquid water path in the parameterised cloud.

As often with atmospheric models, the main difficulty arise when coupling processes occurring on different spatial and temporal scales. We are aware that the hybrid approach assuming two linked compartments is not perhaps the ideal way to simulate wet removal. However, given the Lagrangian framework adopted in this study we think that this approach is sufficiently well suited to answer the overarching questions posed.

Section 2.4 in the original MS outline this approach in detail.

*(4)  P.3, ln. 78-79: The sentence "Therefore, the total . . . no dynamic feedbacks on the aerosol-cloud interaction" needs additional explanation. Aerosol-cloud interaction has various feedbacks such as cloud albedo, cloud lifetime, and thermodynamic and microphysical invigorations. Which feedback processes can cause underestimation of total wet deposition and how?*

We clarify this by adding following paragraph after P.3 ln. 78-79:

"This dynamic feedback can be viewed from two different angles linking to competitive growth during cloud formation on the one hand, and condensation growth on the other hand. As wet deposition to a substantial degree occurs through nucleation scavenging, parts of the available CCN's are removed. If the PNSD remains static, a second cloud cycle will tend to result in activation starting at a lower size range, i.e. the activation and subsequent removal "eats" its way through the distribution taking chunks away from right-to-left with increasingly lower activation radius as a result. This will make smaller and smaller particles available for in-cloud scavenging and removal. Now, if the activation diameter instead would be fixed, and for simplicity assuming it to be 100nm, once all 100nm particles are removed no more in-cloud scavenging can occur. This of course is unrealistic as it is well established that activation is controlled by competitive growth, and the lower number of large particles, the smaller activation radius will result for a given set of conditions. Hence, the cloud activation radius and removal will be adjusted based on the previous removal events. The second link is when particles are grown due to condensation growth (and to lesser extent coagulation). These processes bring particles that otherwise would be too small to be activated into a size range where they in fact may become actual CCN's. Studying this dynamical coupling (or feedback) between growth, removal and cloud droplet activation is one of the main targets of this study. "

**(5) Comment:**

*P.3, ln. 85: "these Eulerian approaches including advanced aerosol dynamics are however not easily adapted to a Lagrangian framework." It seems not very difficult. There exists Lagrangian – Eulerian hybrid methods. Recently, Danielache et al. (2019) (https://doi.org/10.2343/geochemj.2.0542) coupled Eulerian chemical processes to a LPDM framework, although it does not seem to couple aerosol dynamics processes to the radio-Cs modeling, as done by the current manuscript.*

**Answer:**

We acknowledge this fact by including the suggested reference in the MS and replacing P.3 ln. 85: *"For emergency preparedness where fast models are essential these Eulerian approaches including advanced aerosol dynamics are however not easily adapted to a Lagrangian framework."*

with:

"There are examples of Eulerian-Lagrangian hybrid models, for example in (Danielache et al., 2019) where the Eulerian part calculated the chemical transformation and the Lagrangian was used for transport. For emergency preparedness fast models are essential. To our knowledge there has been no solution presented providing sufficiently fast and detailed aerosol dynamics representations within the strictly Lagrangian framework applied in LPDM's used for dispersion simulations of radionuclides"

**(6) Comment:**

*P.4, 2. Method: Horizontal and vertical grid resolutions are missing. How they considered vertical motion of trajectory in HYSPLIT is missing. Brief descriptions of CALM may also be needed such as emission inventories (anthropogenic, biomass burning, biogenic, and volcanic emissions), chemical reactions (gas, aerosol and aqueous phase reactions), and aerosol representations (modal or sectional) used. It may help to grasp how their aerosol dynamics parameters are properly set.*

In Section 2.1 resolution information has been added in the sentence on P.5 ln. 131 ending *with "… using fields from the meteorological model GDAS (NOAA 2019)."* It has been replaced by:

"using fields from the meteorological model GDAS (NOAA 2019) with a geographical resolution of 1 degree in latitude and longitude and temporal resolution of 3 hours."

Section 2.4, l 209-237 in original MS provide a description of the cloud activation and wet removal scheme. We agree that information about the resolution could be introduced into the text. On line 214 following sentence "…coinciding with that of the air parcel." we added:

"The vertical resolution of the meteorological model used (GDAS 1 degree) is, starting from bottom-up at 1000hPa with 25hPa resolution up to 900hPa, and then 50hPa resolution up to levels relevant for the current simulations. The vertical resolution is 1 degree. "

**(7) Comment:**

*P.5, ln. 134: "gap" -> "missing data" may be better.*

**Answer:**

Agreed and changed accordingly

**(8) Comment:**

*P.6, ln. 161-162: "super-micrometre size range is however not included in this study". Aerosol dynamics is less important for coarse mode particles or chemical evolution may enhance the deposition efficiency of coarse mode particles. Neglecting the presence of coarse mode*

*particles in this study may overestimate the impacts of aerosol dynamics processes, which should be noted. If the surface area of coarse mode particles is larger than that of ultrafine particles, gaseous Cs are more condensed to the surfaces of coarse mode particles.*

**Answer:**
Regarding the coarse mode, the referee is right that initial attachment of nuclides to coarse mode particles potentially can give very different results due to rapid sedimentation. This will be, on average, highly dependent on season and geographic location of model initialization as coarse mode particles show high variability in space and time due to the short residence time. In designing the study we did however target the submicron fraction since this is the size range where aerosol dynamical processes play their biggest roles. We acknowledge in the revised MS that including also coarse mode to yet unknown degree can substantially change the result.

After P.6, ln. 161-162 we also added the following:

"Doing so, we are aware that the omission of coarse mode potentially could introduce significant deviation from current results. This is especially true for occasions where the surface area is completely dominated by super-micron particles. Being prone to rapid dry deposition through sedimentation, similar situations could result in comparably fast removal of attached radionuclides. "

> **(9) Comment:**
> *P.6, ln. 170: "Matlab function fmincon.m" Use mathematical expression, rather than a function name of a software.*

On P.6 L170-107 the following sentence has been replaced:

*"The fitting was performed using the Matlab function fmincon.m, that performs a constrained best fit of the three modes to each size distribution."*

by:

"The best fit was found through solving a constrained minimization problem that gives the optimal distribution into three distinct log-normal modes under the constraint of non-negative weights"

> **(10) Comment:**
> *P. 7, Fig. 2 Units of dN/dlogDp and dS/dlogDp may be missing.*

**Answer:**
In Figure 2 *dN/dlogDp* and *dS/dlogDp* are normalised to maximum concentration (=1) which has been clarified with an addition to the Figure 2 capture. P.7 ln. 170 has been replaced from the original version:

*"Figure 2: Annual development of initial number size distribution (left column) and Initial Surface Size Distribution (right column). Melpitz 2008 (top row), Neo 2010 (second row) and Zeppelin 2012 (third row). Three modal fit calculated from measurements at each station. The bottom row holds total number concentration (left) and total surface area (right)."*

to:

"Figure 2: Seasonal development of initial PNSD, normalised to the maximum in the bin with the highest concentration (left column) and Initial Surface Size Distribution, normalised to the maximum in the bin with the largest surface area (right column). Melpitz 2008 (top row), Neo 2010 (second row) and Zeppelin 2012 (third row). Three modal fit calculated from measurements at each station. The bottom row holds total number concentration (left) and total surface area (right)."

**(11) Comment:**
*P. 7, ln. 180-181: The sentence "the internal mix of the particles into three different chemical groups, ammonium bisulfate, condensable and partially water-soluble organic vapour and non-condensable insoluble compounds" is awkward. First, "vapour" is not "particles". What is "non-condensable"? Either highly volatile gas or non-volatile particles can be both "non-condensable". Where is ammonium nitrate and ammonium sulfate? Same questions for Table 1 in P. 10.*

**Answer:**
Regarding *"vapour"*: this is a typo and has been changed to "compound" wherever motivated i.e. P.7, ln. 181, on P. 10 in Table 1,

Regarding "non-condensable": This refers to particle mass that only can be added through primary emissions. For example, BC. The notation "non-condensable" serves to highlight the fact that these species does not form through gas-particle conversion. Neither can these compounds evaporate. As we only consider BC to have this properties in the current model (mineral dust disregarded) we choose to replace "non-condensable" with "primarily emitted species" and "Insoluble organic vapour" with "insoluble organic compounds".

This has been done in Table 1, P.10 and on P.7 ln. 181 where:

*"… and non-condensable insoluble compounds."*

has been replaced with:
"… and primarily emitted insoluble species."

Regarding ammonium nitrate: We do not include nitrate chemistry.

Regarding $H_2SO_4$: The aerosol condensation scheme is largely identical as the one used in the parent aerosol dynamical model UHMA (Korhonen et al., 2004). In solving the dynamics, we have used the hybrid approach outlined, where the core particles are mapped on a fixed sectional grid, and the liquid phase is determined from equilibrium between liquid and gaseous phase of water vapor and ammonia and associated core constituents. This wet radius is traced using a moving centre approach. This means that in the model, the wet radius represents the partial or full neutralization of $H_2SO_4$, depending on the balance between available $H_2SO_4$, $NH_3$ and water vapour. This implicitly considers the effect of neutralization of sulfuric acid in the particle phase.

**(12) Comment:**
*P. 8, ln. 220-221: What is "column precipitation intensity"? How about the unit?*

**Answer:**

For clarification we have removed the word "column" in the parenthesis and added clarification on P.8 ln. 119 changing it from:

*"given as column precipitation at ground level [mm hr-1])"*

To:

"(given as precipitation at ground level [mm hr-1] representing the all precipitation from the column above ground)"

We also removed the "column precipitation intensity" in the sentence on P. 8, ln. 220-221 since it was erroneously formulated:

*"The fraction removed per mm of precipitation is scaled to the calculated liquid water path (LWP) [g m-2] and column precipitation intensity."*

To:
"The fraction removed per mm of precipitation is scaled to the calculated liquid water path (LWP) [g m-2]."

> *(13) Comment:*
> *P.8, ln. 225-226: "The effect of ice components in clouds is neglected, which may overestimate the scavenging efficiency". Why not "underestimate"? Do you assume that the presence of solid hydrometeors delays the scavenging of aerosols? Why?*

**Answer:**

The reason for saying overestimate is based on the way wet removal is parameterised in the model. In the current model version precipitation removal is calculated based on the number of aerosol particles activated into cloud droplets. The cloud module calculates droplet growth by condensation. When adjusting for the effect of wet removal, we simplify the approach by assuming that the droplets all are of same size once activated, and therefor contribute equally to generation of precipitation. The removal is then calculated from precipitation intensity, LWP and cloud fraction. In a case where hydrometeors of ice would be introduced, the result would be growth of ice particles on expense of the liquid droplets due to Bergeron processes. This would reduce the formation of precipitation sized particles resulting from pure condensation growth, and substantially alter the cloud droplet/ice crystal distribution. Hence, this would (as far as we can judge from our parameterization) create lesser amount of scavenged particles per formed precipitation sized droplet. We do however admit that whether mixed phase clouds do or do not enhance scavenging and removal is far more complex than allowing for sweeping statements as currently present in the MS, and we agree that our claim is unsupported. We cannot in fact determine for sure what the net effect will be in our simulated cases and stand corrected. It is also acknowledged that the treatment of mixed phased clouds, although difficult, certainly merits a second take regarding potentially relevant effects it may have in transport and deposition of radionuclides.

We rephrase as follows:

"The effect of ice components in clouds is neglected, which may alter the scavenging efficiency and ultimately the wet removal rate."

*(14) Comment:*
> *P. 11, Result -> 3. Results*

**Answer:**
Agreed and changed accordingly

*(15) Comment:*
> *P. 15, Fig. 6: It is not clear to the reviewer why the difference between purple and green are very large, whereas the difference between yellow and blue are negligibly small. It looks as if "aerosol dynamics are very important when the simple deposition schemes are applied". To my understanding, the simple deposition schemes are independent on aerosol sizes, and thus the difference between purple and green should be even smaller than the difference between yellow and blue.*

**Answer:**
The "simple deposition schemes" used in the study are still size dependant as well as the below cloud scavenging scheme that has not been the focus in the study. The air/cloud interaction scheme with FAS uses a Fixed Activation Size and it is therefore sensitive to when the size is larger than the chosen activation size. This in turn makes particle growth by condensation and coagulation give a great impact on the result since in many simulations particles grow above the activation size and gets washed out. This creates the differences between the purple and green line on P.15, Fig. 6. The same can also be seen on P. 17 Fig. 7 with the same colours on the lines.

To make this clearer in the MS on P.21, ln.504 the following sentences has been changed:

*"The fixed activation size method, FAS, describes the creation of cloud droplets that might lead to wet scavenging of the aerosols. This is a parameterization closer to the physical processes than a scavenging coefficient. What is missing compared to what goes on in a cloud is the coupling between activated particles and remaining particles for future activation."*

Into:

"The fixed activation size method, FAS, describes the creation of cloud droplets that might lead to wet scavenging of the aerosols which makes it more physically credible compared to a scavenging coefficient. The FAS method includes where in the column above and below the box cloud formation and precipitation occurs whereas the scavenging coefficient only correlates ground level precipitation to the wet deposition with no height dependency. The coupling between activated particles and remaining particles for future activation is however neglected in the FAS parameterization."

At the end of this paragraph a sentence has been added after P.21, ln. 514:

"It makes this scheme sensitive for the choice of fixed activation size and calculating this from current meteorological conditions and the current PNSD would be preferable."

*(16) Comment:*
> *P. 16, ln. 388: "cloud formation is buffered by newly formed particles" does not make sense. Which buffer system (Stevens and Feingold, Nature, 2009) do the authors indicate? The newly formed particles decrease the maximum supersaturation of air parcel to form cloud?*

**Answer:**
This connects back to response outlined under specific comment 4 and the sentence on P.16 ln. 387-388:

"Thus, when the initial precipitation events have removed the bulk of Cs-containing aerosols, the remaining Cs-containing particles are few and cloud formation is buffered by newly formed particles."

Has been changed to:

"Thus, when the initial precipitation events have removed the bulk of Cs-containing aerosols, the remaining $^{137}$Cs-containing particles are few and cloud droplet activation takes place on newly formed particles without $^{137}$Cs."

**(17) Comment:**
*P. 16, ln. 401: "Melpitz and ." and what?*

**Answer:**
It should say "Melpitz and Neo". "Neo" is missing and it has been added.

**(18) Comment:**
*P. 19, Discussion -> 4. Discussion*

**Answer:**
Agreed and changed accordingly